# Modeling signaling-dependent pluripotency with Boolean logic to predict cell fate transitions

Ayako Yachie-Kinoshita[1,2,3], Kento Onishi[1,2], Joel Ostblom[1,2] iD, Matthew A Langley[1,2], Eszter Posfai[4], Janet Rossant[4] & Peter W Zandstra[1,2,5,6,*] iD

## Abstract

Pluripotent stem cells (PSCs) exist in multiple stable states, each with specific cellular properties and molecular signatures. The mechanisms that maintain pluripotency, or that cause its destabilization to initiate development, are complex and incompletely understood. We have developed a model to predict stabilized PSC gene regulatory network (GRN) states in response to input signals. Our strategy used random asynchronous Boolean simulations (R-ABS) to simulate single-cell fate transitions and strongly connected components (SCCs) strategy to represent population heterogeneity. This framework was applied to a reverse-engineered and curated core GRN for mouse embryonic stem cells (mESCs) and used to simulate cellular responses to combinations of five signaling pathways. Our simulations predicted experimentally verified cell population compositions and input signal combinations controlling specific cell fate transitions. Extending the model to PSC differentiation, we predicted a combination of signaling activators and inhibitors that efficiently and robustly generated a Cdx2$^+$Oct4$^-$ cells from naïve mESCs. Overall, this platform provides new strategies to simulate cell fate transitions and the heterogeneity that typically occurs during development and differentiation.

**Keywords** asynchronous Boolean simulation; embryonic stem cell; gene regulatory network; heterogeneity; pluripotency

**Subject Categories** Development & Differentiation; Network Biology; Stem Cells

**Mol Syst Biol. (2018) 14: e7952**

## Introduction

Single-cell-level heterogeneity in gene expression is common in pluripotent stem cells [PSCs; MacArthur et al, 2012; and indeed other stem cell types (Gupta et al, 2011)]. There are two scenarios from which this heterogeneity emerges. Either different closely related cell types co-exist or individual cells transition dynamically between different cell states (dynamic heterogeneity; Hoppe et al, 2014; Miyanari & Torres-Padilla, 2012; Schroeder, 2011). Regardless of its origin, heterogeneity can result in families of gene regulatory networks (GRNs), each with potentially unique responsiveness to endogenous or exogenous perturbations (Eldar & Elowitz, 2010; Rompolas et al, 2013; Singer et al, 2014). One manifestation of this is that different subpopulations of cells may have higher probabilities of generating specific types of differentiated cells following treatment with differentiation-inducing ligands (Chambers et al, 2007; Toyooka et al, 2008).

Distinct PSCs and their associated GRNs appear to be stabilized through extrinsic signals (or signal modifiers; Ng & Surani, 2011), which are typically either supplemented into the medium or endogenously produced (Davey & Zandstra, 2006; Moledina et al, 2012). For example, mouse embryonic stem cells (mESCs) will transition into epiblast stem cells (EpiSCs) if LIF and BMP4 in the medium are replaced with bFGF and Activin A. (Guo et al, 2009; Onishi et al, 2012). Additionally, dual small molecule inhibition of MAPK/ERK kinase (MEK) and glycogen synthase kinase-3β (GSK3β; referred to as the 2i condition) yields mESCs in a naïve/ground state of pluripotency, a state which closely resembles the early, pre-implantation stage epiblast (Tesar et al, 2007; Nichols et al, 2009; Evans, 2011). Notably, GRNs do not serve only as responsive elements to external stimuli, but also as stimulus sources themselves via autocrine/paracrine signaling, resulting in combined endogenous/exogenous feedback loops (Davey & Zandstra, 2006) that influence cell fate transition probabilities.

Here, we hypothesize that PSCs transition between heterogeneous cell states under the constraint of signaling inputs. We describe a simulation framework that depicts each PSC subpopulation as a compilation of related heterogeneous gene expression profiles which emerge depending on the given signaling inputs. This computational framework uses an mESC-GRN consisting of 29 key genes to simulate the regulation of Oct4, Sox2, and Nanog (as well as other mESC-associated genes) as a function of signaling inputs.

1   Institute of Biomaterials and Biomedical Engineering, University of Toronto, Toronto, ON, Canada
2   The Donnelly Centre, University of Toronto, Toronto, ON, Canada
3   The Systems Biology Institute, Minato, Tokyo, Japan
4   Program in Developmental and Stem Cell Biology, Hospital for Sick Children Research Institute, Toronto, ON, Canada
5   Department of Chemical Engineering and Applied Chemistry, University of Toronto, Toronto, ON, Canada
6   Medicine by Design, A Canada First Research Excellence Program at the University of Toronto, Toronto, ON, Canada
    *Corresponding author. Tel: +1 604 822 4838; E-mail: Peter.Zandstra@UBC.ca

Existing Boolean models that simulate the regulation of PSC-GRNs treat each subpopulation as a discrete, steady-state gene expression profile (i.e., attractor) derived from a unique or randomly set initial profile after rounds of Boolean updates, where genes are toggled on and off to satisfy the Boolean logic functions that make up the GRN (Dunn *et al*, 2014; Xu *et al*, 2014; Okawa & del Sol, 2015). While the steady-state attractor approach simulates the presence of different cell states (i.e., subpopulations) within a total PSC population, it does not simulate gene expression variability within each PSC subpopulation, nor capture the single-cell variability (MacArthur *et al*, 2012; Xu *et al*, 2014) and subpopulation dynamics (Kalmar *et al*, 2009; Filipczyk *et al*, 2015) that has been observed in single-cell transcriptome data. We therefore aimed to group closely related gene expression profiles into constructs that enabled the prediction of subpopulation composition. To do this, we used a random asynchronous Boolean simulation (R-ABS) strategy. While in a synchronous Boolean paradigm, all genes in the GRN toggle simultaneously to produce each condition, R-ABS randomly picks a subset of genes at each time step and toggles individual genes asynchronously, resulting in a wider catalogue of transitional expression profiles (Di Paolo, 2001). We employed random asynchronous updates assuming uniform average time delays on every gene because this method not only accurately reflects the various biological observations such as changes in cell compositions, but can also represent both rhythmic and non-rhythmic phenomena (Di Paolo, 2001). Instead of depicting subpopulations as steady-state attractors, we hypothesized PSC states in dynamic heterogeneity are recapitulated as strongly connected components (SCCs), where all gene expression profiles in the set can transition into each other (Fig EV1a). Uniquely, this methodology allows the simulation outputs to be quantitatively compared with experimental observations from both single cell- and population-level experiments (Fig EV1b). In addition, our model also includes feedback loops from the GRN to signaling pathway components, thus allowing for the exploration of a broader and more nuanced array of GRN outputs such as the exit from pluripotency as a consequence of the activation and inhibition of different combinations of five major pluripotency-related signaling pathways (LIF/pStat3, Wnt/β-catenin, Bmp4/pSmad1/5/8, Activin A/pSmad2/3, and bFGF/pERK). Taken together, our strategy represents a new platform, capable of simulating cell fate transitions and heterogeneity, that should have broad applicability to many different biological systems.

# Results

## Simulation framework for PSCs

Our simulation framework took advantage of two key strategies. The first is the R-ABS strategy, where Boolean updates are performed asynchronously on randomly selected nodes (genes) in each simulation update, such that each simulated gene expression state can transition to multiple possible successor states (Albert *et al*, 2008; Garg *et al*, 2008). A profile transition graph, the accumulation of the transitions between unique expression profiles, is conceptually analogous to transitions between single-cell states and is derived from an iterative R-ABS. We believe that a binarized representation of gene expression, which is a common simplification for Boolean-based

simulations, is relevant at the single-cell level given the accumulated observations of bimodal distributions in single-cell gene expression profiles in mESCs (MacArthur *et al*, 2012; Xu *et al*, 2014) and in other cell types (Shalek *et al*, 2013). The relative transition frequencies from one expression profile to its successor profiles can be calculated by counting the individual transitions from the source to the target, which in turn determines the probability of traversing of each profile.

After generating the profile transition graph with R-ABS, the second key element of our approach is to use SCCs to group unique expression profiles. An SCC is defined as a subset of expression profiles where every profile is capable of transitioning into all other profiles in the subset and returning to the original profile over an indefinite number of Boolean updates. This is analogous to a sustained PSC population containing multiple transitioning subpopulations (Bao *et al*, 2009). In the context of population-level PSC state transitions, SCCs represent a dynamically stabilized population as a cluster of heterogeneous single-cell profiles where each transition state can give rise to any of the other states within the SCC. In this study, we considered an SCC as PSC subpopulation which fulfills two criteria: the number of unique (heterogeneous) profiles in the SCC and the stability of the SCC are above thresholds. The model predicts the emergence of subpopulations (SCCs) in response to different input conditions (Fig 1A and B). The gene expression level for any given gene within a particular SCC is predicted by multiplying the sum of expression profile probabilities where the gene is present (ON) by the sum of probabilities of all the profiles in the SCC (Fig 1C upper panel). The expression probabilities are then multiplied with the probabilities of remaining within an SCC (i.e., 1 minus the outgoing transition probability; Fig 1C lower panel, see Appendix Section 1). The population-averaged gene expression level is thereby calculated by taking the sum of these values within each SCC, which is weighted by the proportion of subpopulations (the number of unique profiles in each SCC).

## Mouse ESC-GRN construction

Next, we applied the proposed simulation framework to mESC-GRN. To build the model, we first selected 14 pluripotency-associated genes [Oct4, Sox2, Nanog, Klf4, c-Myc, Esrrb, Tbx3, Klf2, Gbx2, Jarid2, Mycn, Lrh1, Pecam1, and Rex1(Zfp42)] based on prior knowledge (see Appendix Sections 2 and 3 for details on GRN reconstruction; De Los Angeles *et al*, 2015; Kim *et al*, 2008). We also included key lineage specifiers (Tcf3, Cdx2, Gata6, Gcnf), genes known to drive the exit from pluripotency (Chickarmane & Peterson, 2008; Tam *et al*, 2008). For computational efficiency, we aggregated EpiSC-enriched transcription factors (TFs)—Fgf5, Eomes, Otx2, and Brachyury (T)—into a single component in the model termed EpiSC-enriched transcription factors (EpiTFs; Bao *et al*, 2009; Guo *et al*, 2009). We then specified regulatory relationships among the genes by manual curation, including 19 regulations encompassing double-positive or double-negative regulatory circuits and known self-activations for seven genes (Appendix Table S1B). We also introduced Dnmt3b to the model as an epigenetic switch regulator of isoforms of Oct4 (Appendix Section 3-4).

We next set the key signaling pathways (LIF/pStat3, Wnt/β-catenin, Bmp4/pSmad1/5/8, Activin A/pSmad2/3, bFGF/pERK, and PI3K) as consequential effects of gene ON/OFF states by extending

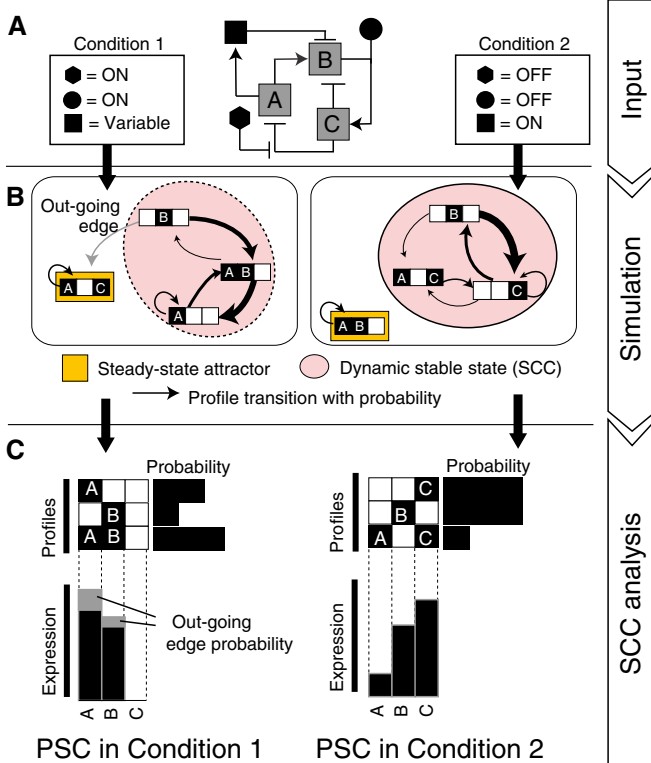

**Figure 1.   Strategy for modeling and simulation of stabilized PSCs.**

A   Experimental conditions can be set as simulation inputs by defining the model variables (signal components with black-filled symbols and genes A, B, and C) as either continuously ON, OFF, or variable.

B   Random ABS generates a directed transition graph where binary gene expression profiles are graph nodes and possible transitions from individual profiles are edges with a certain probability. PSC populations are assumed to be stabilized as a group of heterogeneous profiles, which is defined as an SCC.

C   Weighted, subpopulation-averaged gene expression and signaling activity of a particular SCC are calculated based on transition probabilities and the binary state of each model component in each heterogeneous profile.

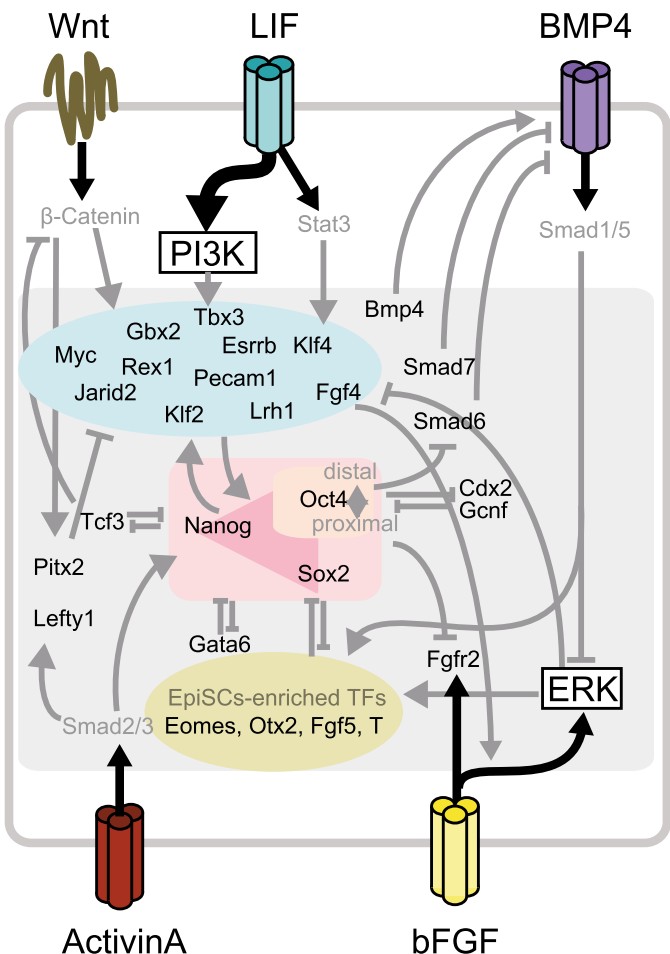

**Figure 2.   A schematic of the defined PSC gene/signal regulatory network model**.

the gene list to link to signaling activities (Fgf4, Fgfr2, Bmp4, and Activin A/Nodal) by manual curation. For example, to model FGF activity and downstream MEK/ERK activation, we included stimulus sources (bFGF or Fgf4) and receptor availability (Fgfr2) in our GRN. To complete signal pathway integration, we then defined regulatory edges from cytokine/receptor-level signaling to their downstream effectors and feedback from genes to relevant signaling activities (Fig 2, Table 1, and Appendix Section 3-3). Importantly, signal activity is not updated asynchronously in our model. Instead, it is calculated at every simulation step based on the ON/OFF states of the genes and their update rules (Appendix Table S2). This simu-lates signaling events as more rapidly occurring and deterministic than gene regulation.

Previous studies have reverse-engineered the PSC-GRN to eluci-date critical regulatory relationships (De Cegli *et al*, 2013; Kushwaha *et al*, 2015). To mitigate biases that could arise from manual cura-tion, we expanded the model to include potential gene regulations inferred from publicly available mESC gene expression data. We performed refined graphical Gaussian modeling (GGM; Ma *et al*,

2007) to infer direct connectedness among genes using a collection of 1,295 publicly available microarray expression datasets for mESCs (Appendix Section 2 for the details and the justifications of the method). This resulted in a network of 29 genes including genes which are predicted to be highly correlated with the previously noted core pluripotency genes (Lefty1, Pitx2, Dusp6, Smad6, and Smad7) listed in Table 1. The network included 86 inferred pairwise gene regulatory relationships (Appendix Table S1A). Directionality was determined for 76 of these gene pairs by either experimental evidence or gene function annotation. The directionality for the remaining 10 gene pairs was determined by subsequent model selection based on fitting to reported single-cell gene expression frequency. Taken together, manual curation supplemented with reverse engineering-based GRN reconstruction led to an expanded GRN-signaling hybrid model consisting of 29 genes, 105 regulatory interactions between genes, seven signaling pathway activities, and 24 regulations down-stream of the signals (Table 1 and Appendix Table S2).

Boolean logical functions of a target gene define the consequence of the binary states of its regulators with AND, OR, and NOT logic operators. Knowledge of all possible combinations and nesting of the operators significantly increase the number of possible models.

**Table 1.  Boolean definition for genes in the model.**

| Gene | Gene category | Boolean definition |
|---|---|---|
| Activin A/Nodal | Cytokine | (SignalACT or Oct4) and (not Sox2) and (not Lefty1) and (not Gbx2) |
| BMP4 | Cytokine | SignalBMP or Gbx2 or Tbx3 or Myc |
| Dnmt3b | Enzyme | (Mycn or Tcf3 or EpiTFs) and not (Cdx2 or Klf4) |
| EpiTFs | Lineage TF | (SignalBMP or Pitx2 or Dusp6) and (not Cdx2) and not (Klf4 and Sox2) |
| Esrrb | Pluripotency TF | (Klf4 or Klf2 or Nanog or (SignalWNT and (not Tcf3))) and (not EpiTFs) |
| Fgfr2 | Receptor | ((SignalFGF) or Gcnf or Cdx2) and (not Nanog) and (not Oct4) |
| Gata6 | Lineage TF | (Gata6 or SignalERK) and (not Klf2) and (not Nanog) and (not Fgf4) |
| Gbx2 | Pluripotency TF | ((SignalWNT and (not Tcf3)) or SignalLIF) and ((Esrrb or Jarid2) and not (Tbx3)) |
| Gcnf | Lineage TF | (Gata6 or Cdx2) and (not EpiTFs) |
| Jarid2 | Pluripotency TF | Klf4 or Oct4 |
| Klf4 | Pluripotency TF | (SignalLIF or ((Klf2 or Klf4) and Nanog and Esrrb and (Oct4 and Sox2))) and (not EpiTFs) |
| Nanog | Pluripotency TF | (Nanog or SignalACT or (Oct4 and Sox2) or Tbx3 or Lrh1 or Klf4) and not (Tcf3 or Gata6) |
| Oct4 | Pluripotency TF | (((Oct4 and Sox2) or Nanog or Klf2 or Klf4) and not (Cdx2 and Oct4) and (not Dnmt3b or Klf2)) or (((Oct4 and Sox2) or Nanog or Lrh1 or Klf2 or Klf4) and (not Gcnf) and (Dnmt3b and (not Klf2))) |
| Smad6 | Signal antagonist | (SignalBMP or Gata6) and (not Oct4) |
| Smad7 | Signal antagonist | (Oct4 or Nanog or Esrrb or Klf4 or Tbx3) and (not Gbx2) and (not Jarid2) |
| Sox2 | Pluripotency TF | Nanog or (Oct4 and Sox2) |
| Tcf3 | Lineage TF | (Nanog or Oct4) and (not SignalWNT) |
| Cdx2 | Lineage TF | (SignalBMP or Cdx2) and not (Cdx2 and Oct4) |
| Dusp6 | TF | SignalERK |
| Fgf4 | Pluripotency TF/Cytokine | Esrrb or Nanog or (SignalWNT and (not Tcf3)) |
| Klf2 | Pluripotency TF | ((Sox2 and Klf4) or Mycn) and (not Pitx2) and (not Dusp6) |
| Lefty1 | TF | (SignalACT or (SignalWNT(not Tcf3))) or Mycn or (Oct4 and Sox2)) and (not Jarid2) and (not Fgf4) |
| Lrh1 | Pluripotency TF | (Tbx3 or Klf4 or (Oct4 and Sox2)) and (not Tcf3) |
| Mycn | TF | (Oct4 and Sox2) and (not Nanog) |
| Pitx2 | TF | (SignalACT or (SignalWNT(not Tcf3))) and (not Sox2) and (not Jarid2) |
| Tbx3 | Pluripotency TF | (SignalPI3K or Tbx3) and (Esrrb or Nanog or Klf4) and (not SignalERK) and (not Tcf3) |
| Myc | Pluripotency TF | ((SignalERK or (SignalWNT and (not Tcf3))) or SignalLIF or Gbx2) and (not Nanog) |
| Pecam1 | Pluripotency Marker | (Klf2 or Nanog) and (not EpiTFs) |
| Rex1 | Pluripotency Marker | (Nanog or Sox2 or Lrh1 or Klf2 or Esrrb) and (not EpiTFs) |

Although this extends the capability of the model to describe a variety of regulatory topologies (Dunn et al, 2014; Xu et al, 2014), our focus was on applying a test model to the R-ABS/SCC approach to predict the response of PSC-GRN to a wide range of signaling inputs involving heterogeneity. We used a biologically relevant and widely adopted rule where positive inputs are combined using OR functions and negative inputs are combined using AND functions. This rule states that a target gene will be present when one of its activators is present and concomitantly all of its repressors are absent. Exceptions were made from manual curation for genes whose coded proteins likely make a complex and work synergistically in regulating target gene expression (e.g., Oct4-Sox2 and Oct4-Cdx2). This resulted in 27,648 possible models including the unresolved directionalities of the above-mentioned 10 gene pairs, each of which has a distinct GRN topology. Among these possible models, we selected the top scoring model whose population-averaged gene expression level minimized Euclidean distance from single-cell expression data of mESCs (MacArthur et al, 2012; Kolodziejczyk et al, 2015) in standard culture conditions that contain LIF and fetal bovine serum (LS; Appendix Section 3-5 and 3-6). The full representation of the model is shown in Fig EV2, Table 1, and Appendix Table S2.

## Model recapitulates distinct PSC states

Using our GRN model and simulation strategy, we assessed the ability to predict PSC responses to different input signals. Mouse ESCs in LIF+serum medium (LS) were simulated by setting LIF as continuously ON and allowing other endogenous signaling to undergo state transitions based on the logic functions comprising the network (Fig 3A). Using the LS input rule, we identified only one SCC which

had no outgoing edges. Five distinct steady-state attractors were also identified in LS conditions. The predicted population-averaged gene expression levels of pluripotency-associated transcription factors were comparable with those reported using single-cell RT–PCR in LS conditions (Fig EV3A). Interestingly, the LS model also predicted that Oct4 was likely to co-exist with EpiTFs, while Sox2 showed a strong negative correlation to EpiTFs, an observation consistent with previous reports (Hayashi *et al*, 2008; MacArthur *et al*, 2012; Fig EV3B).

To demonstrate the ability to predict alternate PSC states in response to changes in input signaling, we next performed simulations for EpiSCs (Brons *et al*, 2007; Tesar *et al*, 2007) and naïve mESCs (Ying *et al*, 2008) by changing only the input from LS to bFGF+Activin A (bF+A) or to LIF combined with inhibition of MEK and GSK3β (2iL), respectively (Fig 3A). Simulations of both bF+A and 2iL conditions yielded only one PSC-associated SCC (Fig 3B; see Materials and Methods and Appendix Section 5 for details). Notably, despite the fact that EpiSC gene expression data were not used to construct our generic PSC network, the bF+A simulation predicted expression levels unlike those of mESCs in LS but closely resembling experimental observations for the EpiSC state (Figs 3C and EV3C). Meanwhile, simulations of the 2iL condition did not show significant differences in expression compared to the LS condition, including expression of major pluripotency-supporting factors (Marks *et al*, 2012). This is consistent with the biological observation that LIF is sufficient to maintain mESC-specific gene expression patterns (Smith *et al*, 1988). These data demonstrate that changing model inputs can drive GRN states to those observed in population-level *in vitro* experiments.

We next asked whether direct manipulation of the GRN nodes would lead to shifts between PSC states. This was done by setting individual genes ON (gain of function; GOF) or OFF (LOF), permanently, regardless of their effector states. These simulations predicted Klf4, Nanog, Esrrb, Myc, and Gbx2 as drivers of EpiSC to ESC transition, and Tcf3 to be an inhibitor (Figs 3D and EV3D). These *de novo* results are consistent with previous experimental observations (Guo *et al*, 2009; Hanna *et al*, 2009; Bernemann *et al*, 2011; Festuccia *et al*, 2012; Martello *et al*, 2012; Tai & Ying, 2013; Joo *et al*, 2014). The model also predicted that activating BMP4 while in bF+A conditions (i.e., EpiSC GRN) buoyed Oct4, Sox2, and Nanog (OSN) levels (Fig EV3E), an observation that may explain the positive role of BMP4 in early stages of EpiSC reversion (Bernemann *et al*, 2011). Taken together, these findings demonstrate that our model can be used to predict how manipulating both extrinsic signals and/or endogenous GRN components yields gene regulator network topographies associate with distinct PSC states.

### LIF stabilizes pluripotency while 2i up-regulates OSN

The medium conditions LIF+serum (LS), 2i+LIF (2iL), and 2i without LIF are all sufficient to support stable PSCs (Nichols *et al*, 2009; Wray *et al*, 2011; Martello *et al*, 2012; Yeo *et al*, 2014) and their GRNs (Nichols *et al*, 2009; Dunn *et al*, 2014). Yet, clear morphological and phenotypic differences exist between cells cultured in baseline LS versus those supplemented with 2i (Fig 4A). Therefore, we searched for quantitative metrics from our Boolean GRN simulation approach that could explain this observation. We mathematically defined three metrics: "*pluripotency*", "*susceptibility*", and

"*sustainability*" (Fig 4B and Appendix Section 4) as follows. *Pluripotency* is the population-averaged OSN expression level (sum of Oct4, Sox2, and Nanog levels). *Sustainability* is a score that reflects stability of an SCC in the absence of further perturbation. *Susceptibility* quantifies the difference between an unperturbed SCC and an SCC with a perturbation of a GRN component (see "Calculation of population properties based on SCC" section in Materials and Methods for full formulations). These metrics facilitated quantitative comparisons of GRN properties in the context of dynamically stabilized cell states.

Computationally, the SCCs identified in the 2iL and 2i−L (i.e., 2i *minus* L, i.e., LIF input was set as OFF) conditions had a higher *pluripotency* score than the SCCs identified in the LS condition (Fig 4C-i). This prediction was validated *in vitro* with immunocytochemistry for OCT4, SOX2, and NANOG (Fig EV4A) and is consistent with data from previous reports (Kolodziejczyk *et al*, 2015). Pairs of individual OSN components were more strongly correlated in 2i-containing conditions than LS (Fig EV4B), indicating higher self-sustenance and homogeneity of the core network.

Upon examining *sustainability,* our simulation scored the 2i−L model lower than the 2iL and LS models (Fig 4C-ii). As the sustainability score reflects the ability of a subpopulation in a given condition to maintain itself over time, the prediction infers that the presence of LIF raises the intrinsic stability of the subpopulation. This is consistent with previous observations that LIF does not affect the pluripotency of mESCs in 2i-supplemented conditions, but enhances colony-forming efficiency (Wray *et al*, 2010; Martello *et al*, 2013).

Finally, to measure the *susceptibility* metric upon perturbations to GRN topology *in silico*, we removed individual regulatory relationships from the original model and quantified the resulting change in the population-averaged expression profile. This analysis demonstrated that the GRN in 2i−L was more susceptible to perturbations to GRN topology than in conditions containing LIF (Figs 4C-iii and EV4C). For example, removal of the positive regulatory link from Nanog to Esrrb decreased OSN expression levels in 2i−L but not in LS and 2iL. This indicates that this link lacks the inherent redundancy to sustain OSN levels in the absence of LIF signaling. This finding confirmed the results from Dunn *et al* (2014) that dual LOF of Nanog and Esrrb results in significant loss of pluripotency in 2i−L but not in 2Il. Additionally, our model recapitulated the outcomes of the single- and double-gene LOF studies presented in their report (Dunn *et al*, 2014). Taken together, our simulations suggest that 2i drives PSCs into a naïve state expressing homogeneous levels of OSN, in part by supporting the OSN subnetwork. Furthermore, the addition of LIF to 2i increases sustainability and decreases susceptibility of the overall GRN to perturbations which, potentially by functional redundancy or additive effects (Martello *et al*, 2013; Fig 4B), are predicted to create barriers to the exit from pluripotency.

Based on our quantitative metric-based analysis, we next hypothesized the pluripotency GRN supported by different input conditions (2iL and 2i−L) would be differentially susceptible to exogenous molecular perturbations. Indeed, simulation of all possible signal inputs (Fig 4D) predicted that although 2i-containing conditions (red and blue) give higher overall OSN expression levels than +LIF (orange) or −LIF (black) conditions without 2i, the 2i−L condition has a higher variance of OSN levels ($F = 0.064$, $P$-val $= 8.0e-4$; $F = 0.011$, $P$-val $= 2.2e-4$; $F = 0.136$, $P$-val $= 1.1e-2$ for Oct4, Sox2,

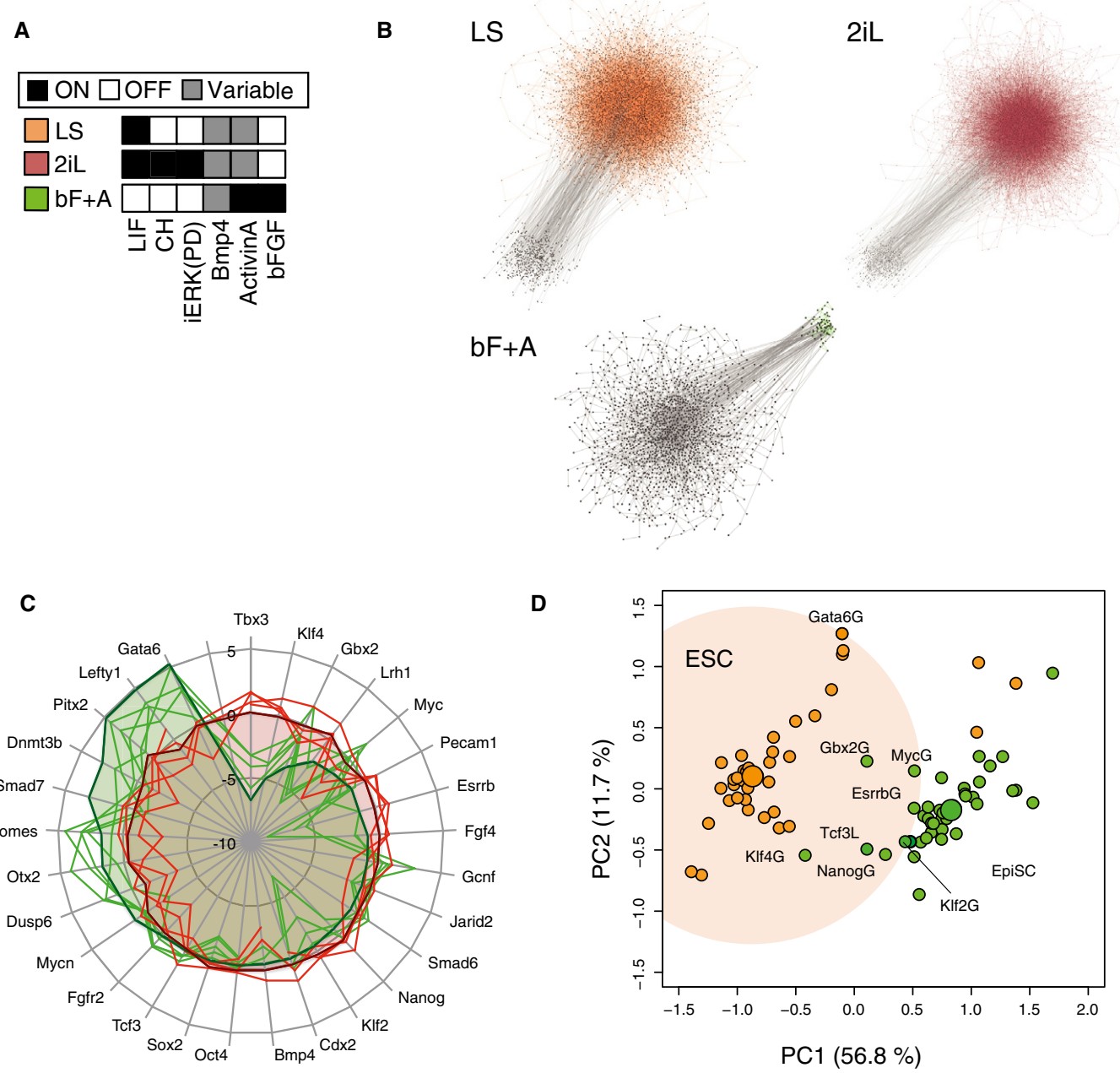

**Figure 3. Simulation recapitulates distinct PSC states.**

A   Simulation inputs for LIF+Serum (LS: orange), 2i+LIF (2iL: red), and bFGF+Activin (bF+A: green) conditions.

B   Condition-dependent pluripotent cell populations correspond to strongly connected components (SCCs) in the state transition graphs of asynchronously updated Boolean models. Gray dots represent unique profiles, and edges represent state transitions among the profiles. Colored edges indicate the transitions within population-specific SCCs. The number of simulations and the number of steps in each simulation were 300-100, 300-100, 300-300 for LS, 2iL, and bF+A condition, respectively.

C   Pinwheel diagram of relative population-averaged expression levels in predicted states (shaded area) under different input conditions (red—mESC conditions, green —EpiSC conditions) recapitulates experimental gene expression data (solid lines) from microarray and RNA-seq studies.

D   *In silico* single gene GOF/LOF analysis of mESCs and EpiSCs was performed by fixing each gene in the GRN as ON or OFF, in either mESC (LS—orange) or EpiSC (bF+A —green) conditions. The calculated gene expression levels following each manipulation were mapped onto principle component analysis (PCA) metrics. The individual gene perturbations that resulted in the changing of overall gene expression of EpiSCs to a more mESC-like one (green dots in orange shaded space) were predicted candidates for driving reversion from EpiSCs to mESCs.

and Nanog, respectively). Notably, OSN levels were predicted to decrease in the 2i−L condition only when combined with high BMP4 and low Activin A/Nodal (2i−L+B−A). To test these

predictions, we measured core pluripotency GRN responses to combinations of four signaling inputs (LIF, BMP, WNT, and Activin A/Nodal), both *in silico* (Fig 4E) and *in vitro* (Fig EV4E). To fully

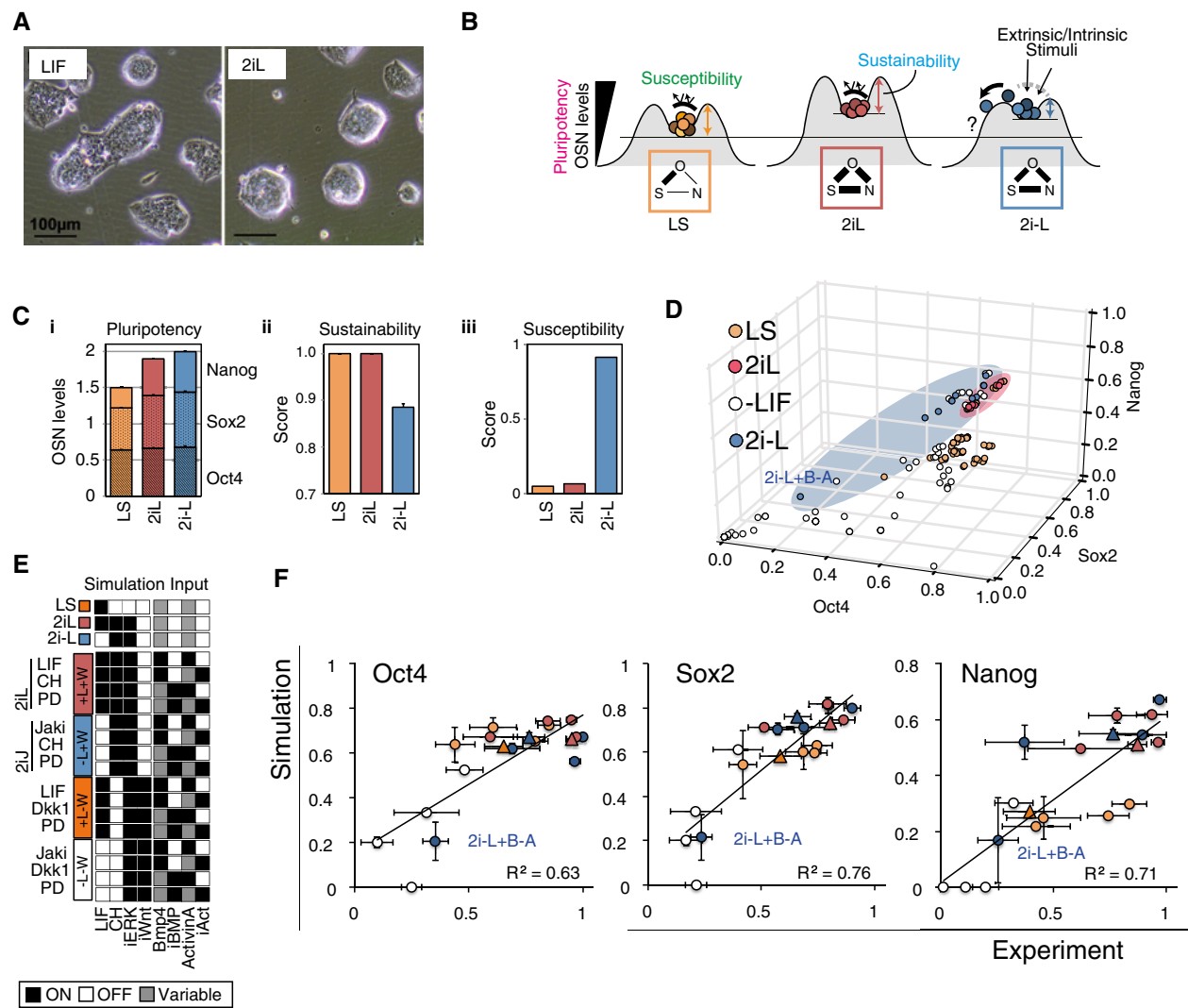

**Figure 4. Dual inhibition (2i) supports the pluripotency core network (OSN), while LIF stabilizes PSCs.**

A   Representative bright-field microscope images of mESC colonies in LIF and 2iL conditions with serum. The 2i condition consists of CHIR99021(CH) and PD0325901(PD).

B   Schematic illustration of the PSC metrics. The frequency of OSN-high cells reflects the population-level pluripotentiality. Sustainability reflects the intrinsic network stability during maintenance of the PSC state in the absence of extrinsic stimuli. Susceptibility measures the change of expression profiles to perturbations such as gene manipulations and signaling inputs and predicts the chance of PSC fate change. The link width among OSN in each condition represents the Pearson's correlations among OSN.

C   (i) Pluripotency level (OSN expression) of each PSC-associated SCC. (ii) Sustainability scores for each PSC-associated SCC. (iii) Susceptibility of gene expression profiles against minimal perturbation to GRN topology was assessed *in silico* by measuring the change of variance in all genes. The error bars represent s.d. of five independent simulations.

D   Predicted population-averaged gene expression levels of OSN in SCCs from all possible combinations of signal inputs (white—without LIF, orange—with LIF, red—with 2iL, and blue—with 2i−L).

E   Four signaling pathways are manipulated in 16 conditions that are divided into four groups based on LIF and Wnt signal manipulations: +L+W (red, 2iL), −L+W (blue, 2iJ), +L−W (orange), and −L−W (white). Note that 2i+JAKi (2iJ) is the *in vitro* counterpart to the *in silico* 2i−L.

F   High content screening results of gene-expressing cell frequency (*x*-axis) and predicted population-averaged expression levels (*y*-axis) of OSN. Each condition is tested under activated and repressed Activin A/Nodal and BMP signals by addition of cytokines or inhibitors (±A±B). The symbol + indicates the addition of cytokines or small molecules that results in activation of the signaling pathway, and the symbol − indicates the addition of inhibitors to pathway activity. Circles are 16 combinatorial signal conditions, and triangles are the three control PSC conditions and are colorized using the same scheme outlined in (E). Experimental data are represented as mean and s.d. of three experiments, each performed in two replicates, and simulation data represent five independent simulations.

Source data are available online for this figure.

recapitulate simulation inputs, all signals that were turned OFF were validated with a corresponding small molecule inhibitor as listed in Appendix Table S4 [e.g., −L in simulations = Janus kinase (JAK)

inhibitor; JAKi (J) in experiments]. Because LIF and WNT contribute to the maintenance of naïve mouse pluripotency (Smith *et al*, 1988; ten Berge *et al*, 2011), we categorized each condition by the

presence or absence of LIF and WNT signaling. Conversely, OSN levels were high for conditions containing LIF or WNT, both in simulated and *in vitro* conditions (Fig EV4F). One notable exception where OSN levels were low even in the presence of WNT was the 2iJ+B−A condition which is supplemented with BMP4 and ALKi, the inhibitor for Activin signaling receptor (Activin receptor-like kinase 4/5/7, ALK4/5/7). Importantly, there was a high degree of correlation in OSN levels between *in silico* simulations and *in vitro* validations for all conditions tested (Fig 4F). In conditions without LIF and with high WNT signaling (2iJ), OSN levels were sustained as high as the conditions with LIF and with low WNT signaling (Fig EV4G). Interestingly, however, susceptibility of 2iJ conditions to perturbation by BMP and Activin signaling increased (Fig EV4G, blue bars). These observations were conserved regardless of the presence of serum (Fig EV4H). Overall, these studies demonstrate that pluripotent cell states maintained under different input signaling conditions are differentially susceptible to destabilization of the pluripotency GRN, with the 2iJ conditions being particularly sensitive to perturbation. Note that a control consensus interactions-only model which excludes the 10 predicted interactions but includes those validated from literature or ChIP-based genome interaction studies did not accurately predict the OSN levels in the various signal combinations in Fig 4F (Appendix Section 5-4).

### More permissive loss of pluripotency from 2i in the absence of LIF

We next set out to characterize the exit of PSCs from pluripotency using the susceptible 2iJ-induced state. We first confirmed that mRNA levels of OSN are decreased in 2iJ+B−A (2i-L+B−A *in silico*) after 2 days of culture in the respective conditions (Fig 5A). Furthermore, we found that the population-averaged gene expression levels of genes typically scored as extraembryonic lineage specifiers (Cdx2 and Gata6) were significantly higher in the 2iJ+B−A condition versus control conditions (2iL or 2iJ/2i−L) both in simulations and in qRT–PCR experiments (Fig 5A). Furthermore, deletion of any combination of two genes from the pluripotency supportive genes (Esrrb, Gbx2, Klf2, and Jarid2) in the model failed to predict the up-regulation of Cdx2 in the 2i−L+B−A condition. Differentiation of naï mESCs to trophoblast stem (TS) cell-like cells occurs upon the forced expression of the trophoblast master regulator Cdx2, through the addition of medium components (Niwa *et al*, 2005; Hayashi *et al*, 2010). Moreover, apparent totipotency from mESCs derived in 2i (Morgani *et al*, 2013) has been reported, and BMP signal activation helps drive trophoblast gene expression from mouse and human primed PSCs (Brons *et al*, 2007; Vallier *et al*, 2009; Bernardo *et al*, 2011). We thus asked whether we could use our increased understanding of pluripotent cell state susceptibility to specifically direct the exit from pluripotency and access gene expression profiles normally reticent to differentiation from mESCs.

Taking advantage of our framework's capacity to predict differentiation trajectories upon exit from pluripotency, we scored individual SCCs and steady-state attractors as candidates for lineage bias based on both high expression of lineage markers and low expression of Oct4. We considered the lineage markers Cdx2 (trophectoderm-associated—TE), Gata4/6 (mesendoderm-associated—ME or primitive endoderm-associated—PE), and EpiTF genes (post-implantation epiblast; Figs 5B and EV5A).

To explore conditions predicted to induce TE-associated genes, we measured Cdx2 protein expression. As Cdx2 can also emerge during primitive streak development (Bernardo *et al*, 2011), we also co-stained with Oct4 and lineage markers Gata4 (endoderm) and Brachyury (mesoderm; Fig 4C). There was a marked increase in the Cdx2 single-positive subpopulation in 2iJ+B−A, but not in the conditions lacking 2i, Jaki, BMP4, or ALKi (Fig 5D). The robust contribution of 2iJ+B−A toward a Cdx2-high state over time was confirmed by the frequency of Cdx2$^+$/Oct4$^-$ cells after extending treatment to 5 days (Fig EV5B). To further investigate this Cdx2$^+$ state, we assayed a supplementary panel of TE-associated markers by flow cytometry and qRT–PCR. We observed that TE-associated genes, such as Trop2 and Hand1, also showed marked increase in expression in 2iJ+B−A relative to controls (Fig 5E). Increased expression of both TE-associated surface markers CD40 and CUB domain-containing protein 1 (CDCP1; Rugg-Gunn *et al*, 2012) was observed in 2iJ+B−A (Figs 5F and EV5C). Importantly, however, RNA-seq and subsequent PCA demonstrated that there exists a separation between mESCs in 2iJ+B−A and trophoblast stem cells (TSCs; Fig 5G). This suggests that the fate transition of mESCs in 2iJ+B−A to a TE-like state is incomplete. Nevertheless, this analysis demonstrates overall consistency between the simulation outputs and the differential expression profiles of mESCs in 2iJ+B−A from those in 2i−L. The significantly up- or down-regulated genes in 2iJ+B−A were highly predictable (Fig 5H), a notable achievement for this complex cell-transition process.

Our analysis thus far demonstrates the strong predictive power of the simulation framework. We next tested whether 2iJ-treated mESCs, which we predicted are in an unstable and signal-responsive pluripotent state, would contribute to developing *in vivo* embryogenesis differently from other pluripotent conditions. We aggregated 2iJ-treated (2d) mESCs to totipotent host embryos (8-cell stage embryos) and allowed endogenous cues to guide differentiation of these cells during pre-implantation development. We noted an increased frequency of 2iJ-treated cells localizing to TE positions in the blastocyst compared to 2iL conditions (Fig 5I, left panel), which was confirmed with different cell line (Morgani *et al*, 2013; Fig EV5D). However, these TE-positioned cells did not express Cdx2 at the time they were assayed *in vivo* and many seemed to have initiated apoptosis (Fig 5I, right panel). These results suggest 2iJ-treated cells are in an altered state of pluripotency, but are not fully competent to undergo trophoblast differentiation *in vivo*, either due to incomplete *in vitro* programing to a TSC like state (such as suggested by our RNA-seq analysis) or due to competing signals received *in vivo*. These differences between 2iJ/2iJ+B−A-treated mESCs and TSCs, as well as the contradictions between simulations and measurements, suggest a requirement for additional TE-lineage specification beyond the Oct4-low/Cdx2-high state.

## Discussion

Pluripotent stem cells represent a powerful platform for the simulation of cell fate transitions (Morris *et al*, 2014). A variety of methods have been used to model pluripotency. Prior models of mESC-GRNs with ordinary differential equations yielded mechanistic insights into cell fate transitions, but included only a small number of

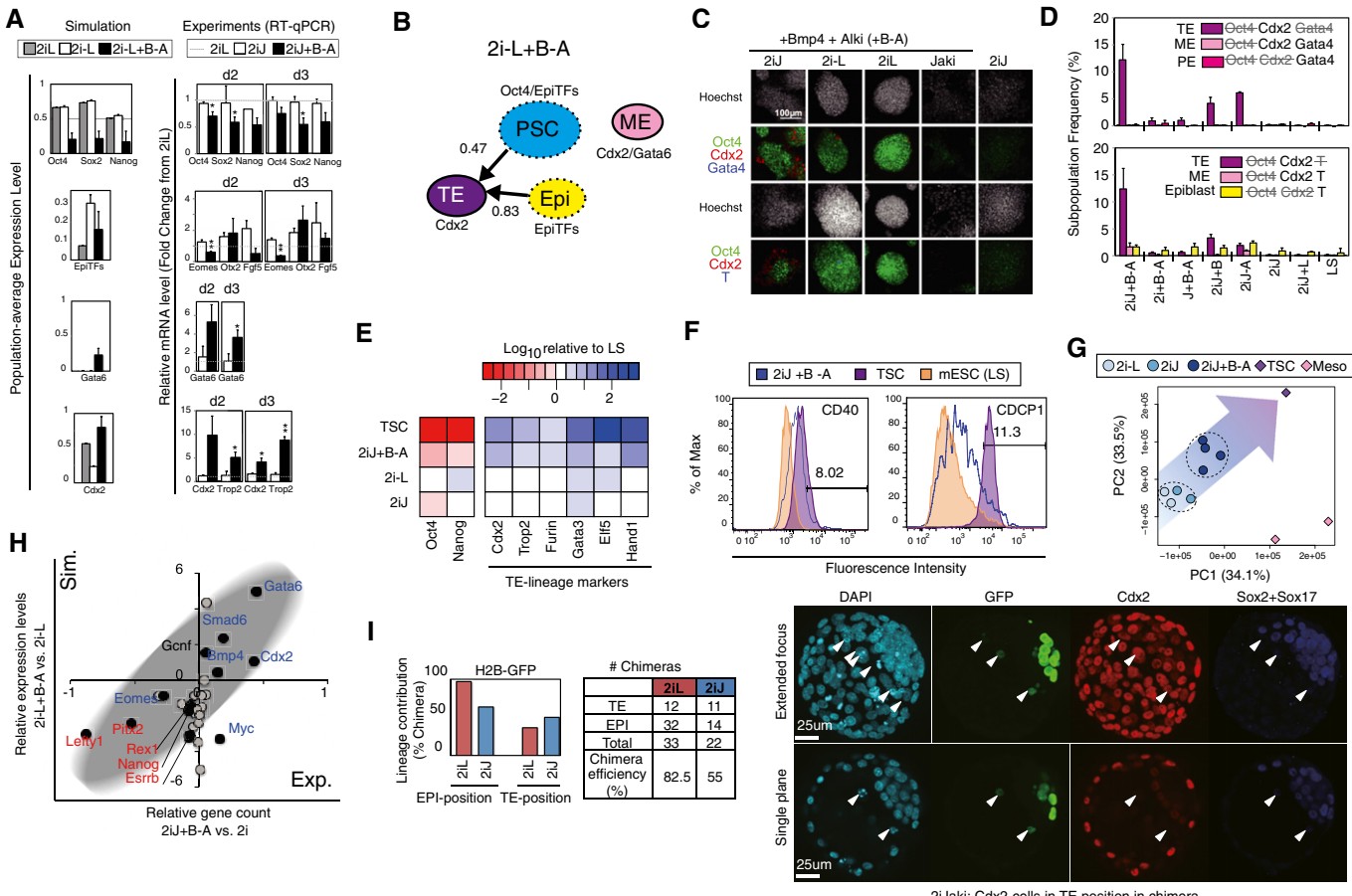

**Figure 5. A culture-induced TE-like subpopulation.**

A  Predicted population-level gene expression levels (left panel) and qRT–PCR-measured relative expressions (right panel; shown in fold-change from the levels of 2iL) of OSN and lineage markers. Data represent the mean and s.e.m. of three or four biological replicates, the differences between 2iJ and 2iJ+B were examined using a two-tailed unpaired Student's *t*-test, and asterisks indicate \*$P < 0.1$ and \*\*$P < 0.05$.

B  *In silico* subpopulation analysis via threshold-based characterization for individual SCCs under the input condition of 2i−L+B−A. Stable grouped profiles enriched as either PSC, TE, ME, PE, or epiblast-like subpopulations were traced in color-coded circles. The circles with solid line indicate SCCs with no outgoing edges (sustainability score = 1.0), and those with dashed line indicate SCCs with lower sustainability.

C  Confocal images of immunostaining of mESCs for Oct4/Cdx2/Gata4 (top) or Oct4/Cdx2/Brachyury (T) (bottom panels) cultured in the given conditions for 2 days.

D  Quantification of frequency of subpopulations that exhibit features of differentiation lineages. Data are means and the error bars represent s.d. of three replicates, and the results were confirmed in two independent studies.

E  Comparison pluripotency and extended TE-lineage gene expression levels in various conditions. Data represent the mean of three replicates.

F  Flow cytometry histograms showing fluorescence intensity of CDCP1 and CD40 in individual samples of mESC in LS, TS, and mESCs cultured in 2iJ+B−A for 2 days. Percentage listed is that of positive cells in the 2iJ+B−A condition.

G  PCA plot of RNA-seq data for the top 40% of genes that show highest variance across all samples. Distinct cell types and conditions are indicated with different colors. Circles include day 2 and day 5 samples for 2i−L and 2iJ conditions, and 2 day two samples, day 5 and 11 for 2iJ+B−A condition. Diamonds indicate stable cell type no culture time defined. Meso indicates mesoderm progenitors.

H  Comparison of predicted gene expression levels and RNA-seq-measured gene counts in 2iJ+B−A relative to 2iJ (equivalent to 2i−L in simulation) for 29 genes involved in the model. The experimental mean relative gene expressions of day 2 samples for the two conditions and the mean relative predicted levels are shown. Black dots indicate genes significantly up- or down-regulated ($P < 0.05$) in three 2iJ+B−A-treated samples compared with two 2iJ-treated samples. Genes in blue are up-regulated in TSC, and those in red are down-regulated in TSC compared with 2iJ-treated samples.

I  Left panel: *In vivo* lineage contribution frequency and chimera efficiency of H2B-GFP mESCs treated with either 2iL or 2iJ in the presence of serum. Lineage contribution efficiencies were calculated as number of chimeras with cells in epiblast (EPI) or TE positions/total number of chimeras. Note that cells scored as "TE position" did not express TE marker Cdx2. Chimera forming efficiency was calculated as number of chimeras/number of total aggregates made. Right panel: Representative images of aggregation chimeras at E4.5. We observed a number of cells in TE positions in chimeras. The appearance of these cells ranged from viable to apoptotic; however, none expressed Cdx2 and thus were not considered as viable, integrated contributions to the TE lineage. The arrowheads indicate the Cdx2-cells in TE position in chimera.

Source data are available online for this figure.

---

network components (Glauche *et al*, 2010; Herberg *et al*, 2014). More recent work has expanded network components in GRN models by moving from ODEs to logical models and has derived

network connectivity from either population-level knock-in/knock-down data (Dunn *et al*, 2014) or single-cell expression data (Xu *et al*, 2014). These models, however, were restricted in their ability

to simulate transcriptional heterogeneity and exit from the pluripotent state, in part due to the absence of signaling inputs to GRNs. To address this limitation, we used expression data from mESCs to examine our GRN framework and to produce a model of pluripotency that, when perturbed *in silico*, could recapitulate the signal-dependent emergence of cell subpopulations observed under analogous *in vitro* and *in vivo* conditions. Additionally, by applying concepts from graph theory to our asynchronous Boolean model of pluripotency, we demonstrated the ability to model both exit from pluripotency and heterogeneity (Karwacki-Neisius *et al*, 2013; O'Malley *et al*, 2013; Marucci *et al*, 2014). Importantly, asynchronous updates from a small set of initial random profiles can give a robust prediction of PSC-GRN profiles, and the predictions for each PSC condition were insensitive to the changes in SCC selection criteria [i.e., thresholds for SCC size and sustainability (Appendix Section 5-3)].

The hierarchical differentiation process of PSCs is often illustrated by trajectories of cells, such as in Waddington's metaphorical landscape (Goldberg *et al*, 2007) in which cells bifurcate into different downstream attractors that reflect differentiated cell types (Yamanaka, 2009; Iovino & Cavalli, 2011). Based on the *pluripotency*, *susceptibility*, and *sustainability* metrics in our model, we propose that 2i conditions support a population of cells at the top of this landscape. Activation of the LIF-mediated JAK-STAT pathway stabilizes cells within the stable local valley of the landscape by reinforcing the *pluripotency* GRN, increases the threshold required to induce differentiation, and participates in shaping the landscape with regard to preferential (embryonic) versus non-preferential (extraembryonic) routes. Conversely, inhibition of JAK-STAT signaling destabilizes the *pluripotency* GRN and allows the expression of TE-lineage genes specifically in response to activation of BMP4 and inhibition of Activin A/Nodal signaling.

Although previous reports have demonstrated the apparent ability of human and mouse primed pluripotent cells to differentiate into TE-like cells upon activation of BMP signaling (Vallier *et al*, 2009; Bernardo *et al*, 2011), these results may be condition- and cell line-dependent and have not been connected to the underlying molecular structure of the pluripotency GRN. To date, there has been no evidence for the ability of cytokines and small molecules to drive TE differentiation from naïve pluripotent cells. A recent report from Morgani *et al* (2013) demonstrated that mESC derived in 2i conditions, especially 2iL, increased the potential to contribute to extraembryonic lineages like TE *in vivo*. However, we observed that active inhibition of JAK-STAT signaling in the presence of 2i, with Bmp4 signal activation and Activin A/Nodal signal inhibition, enhanced the induction of TE-lineage genes due to destabilization of the PSC-GRN in a modified feedback signaling network environment. Our cells may require additional signals [such as Notch and Hippo (Rayon *et al*, 2014)] or TE-specific transcription factors (such as Elf5) that were not used here to complete the cell fate transition to fully functional TE cells. Contexts such as DNA methylation of Elf5 or inclusion of a FGF signaling positive feedback loop between CDX2 and Eomes represent possible areas of study in our model (Ng *et al*, 2008). Moreover, a higher epigenetic barrier may separate the TE-lineage and mESCs even in the hypomethylated ground state in 2i (Cambuli *et al*, 2014).

We anticipate that our graph theory-based Boolean simulation approach, which predicts changes in gene expression of sustained

cell populations in response to signaling inputs, can be broadly applied in studies aimed at understanding the key control nodes triggering cell fate transitions. For example, beyond pluripotency, this strategy could help to predict aberrant cell fate transitions in normal or transformed somatic stem cells. In these systems, the exogenous influences typically described as components of the stem cell niche serve to further broaden regulatory feedback (Qiao *et al*, 2014).

The concept of dynamic heterogeneity that is explored here is experimentally supported by single-cell tracking studies (Singer *et al*, 2014; Filipczyk *et al*, 2015) and has been applied in stochastic ODE models of transcriptional noise (Kalmar *et al*, 2009), as well as in the inference of stem cell fate (Feigelman *et al*, 2016). We assume that heterogeneous populations of stem cells correspond to SCCs in the state transition graphs of asynchronously updated Boolean models. However, empirical validation of this assumption would require live cell tracking of multiple genes and cells. A recently reported *in silico* technique (Moignard *et al*, 2015), which derives GRNs by retracing measured single-cell expression profiles associated with asynchronous Boolean transitions, may possibly be used to evaluate how each SCC reflects inter-cell variability. However, such methods present ongoing technical challenges with respect to thresholding continuous gene expression and the quality of the single-cell expression analysis itself.

To encourage further study using our mESC network and Boolean simulation strategy, we have implemented our modeling framework in *Garuda*. *Garuda* (http://garuda-alliance.org) is an open software platform where bioinformatics tools can be discovered and connected into pipelines with other databases and devices (Ghosh *et al*, 2011). Through this platform, the research community will be able to explore and extend our modeling framework without any software coding requirements. We anticipate future extensions of the model into a spatio-temporal setting could predict self-organizing expression patterns in a system where dynamic stability of state transitions drives dynamic equilibrium. We demonstrated the power of the model to predict the cell fate outcomes of PSCs exposed to complex exogenous signals by employing transition graph analysis of random asynchronous Boolean simulation with the new metrics for the stability of the subpopulations. As the framework is amenable to incorporating other levels of biological control, such as epigenetics or metabolism, it provides great opportunities to test models of heterogeneity at both the genetic and cellular (i.e., tumor) levels (Heng *et al*, 2009; Meacham & Morrison, 2013; Qiao *et al*, 2014).

Taken together, our *in vitro* studies confirm the power of our model to predict cell fate outcomes of PSCs exposed to complex exogenous signals. We demonstrate that the model can reveal new biology between different pluripotent cell states including EpiSCs. We also employ new metrics to demonstrate that mESCs cultured in 2i−L represent a unique cell state that exhibits high OSN levels (pluripotency) but are simultaneously highly responsive to a broad array of differentiation-inducing signals (susceptibility). These metrics represent broadly applicable quantitative strategies for GRN scoring in biological systems. Our system also predicts changes in PSC gene expression associate with TE-like cells, an observation validated by robust single-cell gene and protein expression. Interestingly, our analysis also reveals that typical metrics associated with changes in cell state *in vitro* may not translate to *in vivo*

functionality where additional signals context or maturation steps may be required to fully cross cells across fate barriers.

# Materials and Methods

### Random asynchronous Boolean simulation (R-ABS)

Random asynchronous Boolean simulation (R-ABS) was performed using two assumptions: (i) Every combination of model variables is equally likely to be updated in a given step, and (ii) the state-space is generated with all the transition history of a sufficient number of consecutive steps from a sufficient number of random initial states. R-ABS was performed using the BooleanNet ver.1.2.6 Python package (http://code.google.com/p/booleannet/), with 700 consecutive update steps from each of 700 random initial states per condition. Five independent simulations per condition were performed and consistently resulted in similar population-average expression probabilities.

### Calculation of population properties based on strongly connected components (SCCs)

Strongly connected components (SCCs) are defined as clusters of unique expression profiles wherein all profiles are mutually reachable. SCCs were identified in the profile transition graph generated by R-ABS using the NetworkX ver 1.9 Python package.

For a particular SCC with $n$ unique expression profiles, the transition matrix ($M$) with $n$ rows and columns is defined. Each element ($m_{ij}$) of $M$ in row $i$ and column $j$ holds the value of the edge probability (i.e., accessibility from a source profile $j$ to its target profile $i$) ranging from 0 to 1, which represents the relative transition frequencies from a specific expression profile to one of its target profiles among all transitions from the source profile. The profile probability $v_i$ of profile $i$ indicates the chance that a certain cell resides at profile $i$ in the SCC. The matrix product of $m_{ij}$ and $v_j$ indicates the profile probability ($v_j$) of the source profile $j$, which has a transition path to $i$. The distribution approaches a limiting distribution $\boldsymbol{v}$, where $\boldsymbol{v} = M \times \boldsymbol{v}$ is satisfied. Assuming the cell population is a sum of probabilities of heterogeneous single-cell states (profiles), $\Sigma_n(v_n)$ is equal to 1. Solving $v$ under these constraints gives the principal eigenvector of $M$ that tells us the expected population average when the system reaches a dynamic steady state after a certain number of simulation steps from any profile in the SCC.

For each SCC, we define the transition matrix $M$ with elements $m_{ij}$ as follows. For any source profile $j$ and target profile $i$ within a particular SCC, the probability of transitioning from $j$ to $i$ ($m_{ij}$) can be calculated as the observed frequency of transitions from $j$ to $i$ divided by the observed frequency of transitions from $j$ to any profile in the SCC. Thus:

$$m_{ij} = \frac{f_{ij}}{\sum_{k \in K} f_{ik}},$$

where $K$ is the set of all profiles in the SCC.

Sustainability for a particular SCC indicates the probability of remaining within an SCC, that is, 1—probability of outgoing profile transition from the SCC, reflecting a quantitative measure of

the intrinsic stability of the GRN within the SCC. The sustainability score ($S_{scc}$) is defined as $S_{scc} = 1 - (\Sigma_j(v_j) \times \Sigma_k(v_{jk}))$, ranging from 0 to 1, assuming profile $j$ (inside the SCC) has outgoing edges toward profile $k$ (outside the SCC). The expression frequency ($p$) of model component ($g$) in a particular SCC can be calculated as a summation of all $v$ with ON expression of the gene:

$$p_{g,scc} = S_{scc} \cdot \sum_{j}^{n} v_j \cdot \{1(j_g = ON)|0(j_g = OFF)\},$$

where $j_g$ denotes the binary state of $g$ in the profile $j$. To avoid overestimation of $v$ and to maintain calculation accuracy of population-averaged expression level, we defined the thresholds for SCCs to be considered in the analysis as the number of profiles > 10 and sustainability score > 0.7. As a larger dynamic stable state of PSCs is more likely to exist over time, it will become a larger determinant of population-averaged expression levels. Consequently, population-level gene expression level is calculated by averaging multiple SCCs:

$$p_g = \frac{\sum_{scc} p_{g,scc} \cdot n_{scc}}{\sum_{scc} n_{scc}},$$

where $r$ is the number of SCCs found under the given condition satisfying the predefined threshold, and $n$ is the number of unique profiles involved in the SCC. A small constant value for $p$ ($p$ = 1e-5) was applied where necessary to avoid zero division.

We classified each SCC into one of four lineage identities based on the SCC-averaged expression levels of lineage and pluripotency-associated genes (Cdx2, EpiTFs, Gata6, and Oct4) in the associated SCC. Each SCC and steady-state attractor are classified based on the thresholds on the SCC-averaged expression levels (Cdx2: 0.7, EpiTFs: 0.2, Gata6: 0.5 and Oct4:0.3). High expression of individual lineage markers in separate SCCs is used to classify TE (Cdx2), epiblast (EpiTFs), PE (Gata6), and PSC (Oct4), while ME is classified by high co-expression of Cdx2, EpiTFs, and Gata6 in the same SCC (Morgani *et al*, 2013). The population-averaged subpopulation frequency is calculated proportionally to the size and sustainability of each SCC, as follows:

$$p_A = \frac{n_A \cdot S_A}{\sum_{scc} n_{scc} \cdot S_{scc}},$$

where the number of unique profiles in the SCC $A$ and sustainability of $A$ are indicated as $n_A$ and $S_A$, respectively. The population-averaged gene expression and sustainability scores for simulated pluripotent populations are shown in Dataset EV2.

### GRN inference

We collected mESC expression data on 1,295 genes using the Affymetrix Mouse 430 2.0 Array from the Gene Expression Omnibus (GEO) database at the US National Center for Biotechnology Information (NCBI) and ArrayExpress at the European Bioinformatics Institute (EBI; Dataset EV1). Graphical Gaussian modeling (GGM) was employed to infer direct regulatory networks between gene pairs based on partial correlations. Data from all 45,101 probe sets

were quantile normalized with the R/Bioconductor limma package. The probe sets were then collapsed into 13,879 unique genes by taking mean values of probes annotated to the same gene. For each of the 20,000 iterations, 1,000 genes were randomly sampled for pairwise partial correlation analysis with the GeneNet package in R. The GGM score of each gene pair was defined as the lowest partial correlation for the pair over all iterations satisfying an absolute Pearson correlation > 0.3 and $P < 0.05$. Gene-to-gene links with positive/negative-high GGM scores (> 0.03 or < −0.03) were considered as candidate regulatory edges.

## Model selection

The candidate models were evaluated by comparing the Euclidean distance between predicted population-averaged expression levels and observed frequency of gene-expressing cells in single-cell datasets (MacArthur et al, 2012; Kolodziejczyk et al, 2015). Both the simulation and the experiment were performed in the control LS condition. The mRNA expression levels of single cells were binarized by k-means clustering across all samples with k = 2, from which the frequency of gene-expressing single cells in the population was calculated. The Python SciPy package was used to perform k-means clustering. As there are two sets of single-cell transcriptomic reference profiles, the average values of the frequencies from both datasets were used.

## Cell culture

Maintenance of R1 mouse embryonic stem cells (mESCs) was carried out in serum-containing and feeder-free conditions as described previously (Chang & Zandstra, 2004). Validation of predicted responses to exogenous signaling was performed in serum-containing medium supplemented with combinations of the following cytokines/small molecules: LIF (Millipore ESG1107—10 ng/ml), JAK inhibitor (EMD Millipore 420097—2.0 µsM), BMP4 (R&D Systems 314-BP-010—10 ng/ml), LDN193189 (Reagents Direct 36-F52—0.1 µM), CHIR99021 (Reagents Direct 27-H76—3 µM), Dkk1 (R&D Systems 1765-DK-010—275 ng/ml), bFGF (Peprotech 100-18B—20 ng/ml), PD0325901 (Reagents Direct 39-C68—1 µM), Activin A (R&D Systems 338-AC-050—20 ng/ml), and ALK5 inhibitor II (Enzo Life Sciences ALX-270-445—10 µM and Cedarlane ALX-270-445 for RNA-seq). Trophoblast stem cells (TSCs) were cultured as described previously (Tanaka et al, 1998). Mesoderm progenitor cells were generated from embryoid bodies (EBs) in differentiation medium containing Iscove's modified Dulbecco's medium (IMDM; Thermo Fisher Scientific) and Ham's F-12 nutrient mix (Thermo Fisher Scientific) supplemented with 1× B-27 supplement (Thermo Fisher Scientific), 1× N-2 supplement (Thermo Fisher Scientific), 2 mM Glutamax (Thermo Fisher Scientific), 100 U/ml penicillin–streptomycin (Thermo Fisher Scientific), 0.05% bovine serum albumin (BSA; Wisent), 150 µM monothioglycerol (MTG; Sigma), and 0.5 mM ascorbic acid (Sigma). On day 2, EBs were harvested, dissociated into single cells, and re-aggregated in 100-mm Petri dishes (BD Biosciences) with differentiation medium further supplemented with BMP4 (1 ng/ml), Activin A (2 ng/ml), and Wnt3a (3 ng/ml). Mesoderm progenitors were isolated either before or 24 h after addition of IWP-2 (Reagents Direct 57-G89—2 µM) to each Petri dish on day 3.

## mRNA quantification with qRT–PCR

Primer sequences were obtained from PrimerBank (Spandidos et al, 2010) and are listed in Table 2. Gene expression levels were measured by qRT–PCR as described previously (Onishi et al, 2012). Briefly, cells were lysed and RNA was isolated using the PureLink® RNA Mini Kit (Life Technologies). The RNA was converted to cDNA using SuperScript III Reverse Transcriptase (Life Technologies) and amplified in FastStart SYBR Green Master Mix (Roche) using the 7900HT Fast Real-Time PCR System (Thermo Fisher) with an annealing temperature of 60°C. Each dataset was normalized to β-actin in each condition and then normalized to the control.

## In vitro immunostaining and quantification

Cells were fixed and stained as described previously (Onishi et al, 2012). The following antibodies were used at a 1:200 dilution: Oct4 (BD Biosciences 611203), Oct4 rbIgG1 (Cell Signaling 2840S), Sox2 (R&D Systems MAB2018), Nanog (eBiosciences 14-5761-80), Cdx2 (BioGenex MU392-UC), Gata4 (Santa Cruz Biotechnology sc-1237), and Brachyury (R&D Systems AF2085). Stained cells were quantified using the Cellomics (ThermoFisher) high content screening platform. The frequencies of positive cells for single genes were assessed by counting the single cells whose expression levels are above certain threshold (assessed based on the bimodal distribution of the expression level in LIF+Serum conditions) which is common across each technical replicate (i.e., each plate) including two biological replicates for each condition.

## Flow cytometry

Cells (mESCs and TSCs) were first stained for surface markers CDCP1 (R&D Systems AF4515) and CD40 (BD Biosciences 562846) using antibodies at 1:100 dilutions and assayed using flow

**Table 2.**   Primer sequences used in the study.

| Gene | Forward Primer | Reverse Primer |
|------|----------------|----------------|
| Oct4 | AGTTGGCGTGGAGACTTTGC | CAGGGCTTTCATGTCCTGG |
| Nanog | TTGCTTACAAGGGTCTGCTACT | ACTGGTAGAAGAATCAGGGCT |
| Sox2 | GCTCGCAGACCTACATGAAC | GCCTCGGACTTGACCACAG |
| Otx2 | TATCTAAAGCAACCGCCTTACG | AAGTCCATACCCGAAGTGGTC |
| Eomes | GGCCCCTATGGCTCAAATTCC | CCTGCCCTGTTTGGTGATG |
| Gata6 | GGCAGTGTGAGTGGAGGTG | TGGTACGTTCCGTTCAGCG |
| Gata4 | CCCTACCCAGCCTACATGG | ACATATCGAGATTGGGGTGTCT |
| Trop2 | GTCTGCCAATGTCGGGCAA | GTTGTCCAGTATCGCGTGCT |
| Fgf5 | TGTGTCTCAGGGGATTGTAGG | AGCTGTTTTCTTGGAATCTCTCC |
| Bry (T) | GCTGGATTACATGGTCCCAAG | GGCACTTCAGAAATCGGAGGG |
| b-actin | GAAATCGTGCGTGACATCAAAG | TGTAGTTTCATGGATGCCACAG |
| Furin | TCGGTGACTATTACCACTTCTGG | CTCCTGATACACGTCCCTCTT |
| Gata3 | AAGCTCAGTATCCGCTGACG | GTTTCCGTAGTAGGACGGGAC |
| Elf5 | GACTCCGTAACCCATAGCACC | GCTGAACAGATCGGTCCAAGG |
| Tfap2c | TACCAGCCGCCTCCTTACTT | TCCAGCCCTGAAATATGGGGT |
| Hand1 | GGCAGCTACGCACATCATCA | GCATCGGGACCATAGGCAG |

cytometry (BD LSRFortessa). Cells were also stained with a live/dead stain (LIVE/DEAD Fixable Far Red Dead Cell Stain Kit for fixed cells, 7-AAD for live cells—Life Technologies) and gated for live cells. Final graphs were generated using FlowJo software.

### RNA-seq

The extraction of RNA was conducted using the PureLink RNA Mini Kit (Ambion, Life Technologies, Cat no. 12183018A and 12183025) according to the manufacturer's instructions. Cells were homogenized using a QIAshredder (Qiagen, Cat no. 79654). Cell pellets were frozen at the treatment-specific time points, and RNA was extracted from all pellets at the same time for each analysis. Quality control of total RNA was done on an Agilent Bioanalyzer 2100 RNA Nano chip following Agilent Technologies' recommendation. Next, RNA libraries were sequenced on an Illumina HiSeq 2500 platform using a High Throughput Run Mode flowcell and the V4 sequencing chemistry following Illumina's recommended protocol to generate paired-end reads of 126-bases in length. Reads were trimmed for adapters and a phred33 quality cutoff of 20 using TrimeGalore with cutadapt and mapped to the Ensembl NCBIM37 mouse genome using STAR 2.4.2a. To adjust batch effects between experiments of two distinct days, COMBAT, an R package for Empirical Bayes method (http://statistics.byu.edu/johnson/ComBat/) was utilized.

### Chimera generation and analysis

Joshua Brickmann (Copenhagen) provided the E14 Ju09 HV H2B-Tomato mESCs (Morgani *et al*, 2013) and CAG H2B-eGFP ESCs were derived from mice (Hadjantonakis & Papaioannou, 2004). These mESCs were maintained in mESC medium containing 2i/LIF/serum. Cells were passaged twice with 2iL on mouse embryonic fibroblasts and inactivated with no growth factor medium. Cells were then treated for 48 h with 2i/LIF/serum or 2i/Jaki/serum on mouse embryonic fibroblasts. Chimeric embryos were generated by morula aggregation. Clusters of 5 to 10 mESCs were aggregated with wild-type CD1 morulae and cultured in potassium (K) simplex optimized medium (KSOM; Chemicon) under paraffin oil at 37°C and 5% $CO_2$ until the late blastocyst stage (embryonic day 4.5). Blastocyst embryos were subjected to immunofluorescent staining using anti-Cdx2 (1:600, Abcam ab76541), anti-Gata4 (1:100, Santa Cruz Biotech sc-9053), anti-Sox2 (1:100, R&D Systems AF2018), anti-GFP (1:400, Abcam ab13970), and anti-RFP (1:100, Abcam ab65856) antibodies. Imaging was performed using a Quorum WaveFX spinning disk confocal system and Volocity acquisition software (Perkin Elmer). The frequency of cells localizing to extraembryonic—trophectoderm (TE) positions in the blastocyst was counted. The investigators were not blinded to allocation during outcome assessment, and the experiments were not randomized.

### Data and statistical analysis

We assumed that each well of a culture dish behaves as a biological replicate. No statistical methods were used to predetermine sample sizes. Images including immunostaining experiments shown represent at least three independent runs. Simulation data were derived from five individual runs for the indicated inputs. For the calculation of *P*-value, Wilcoxon exact rank test (R: coin package) was used for comparison of data groups unless otherwise stated. All tests of statistical significance were two-sided.

### Data and software availability

RNA-seq data have been deposited in the Gene Expression Omnibus (GEO) under the accession number of GSE88928.

The simulation framework developed in this study is implemented as a set of downloadable *Garuda* gadgets (http://www.garuda-alliance.org). *Garuda* is an open platform that enables interoperable connections between bioinformatic software, databases, and devices into complete pipelines. We have provided gadgets for the Boolean network simulation including R-ABS (http://50.112.254.186/node/88), the SCC profile calculation (http://50.112.254.186/node/87), and the binarization of gene expression data (http://50.112.254.186/node/86). The Python source code is also available at (https://gitlab.com/stemcellbioengineering/garuda-boolean).

**Expanded View** for this article is available online.

### Acknowledgements

This work is funded by the Canadian Institutes of Health Research (CIHR grant #496640) (P.W.Z.). A.Y. has been supported by CIHR, G-COE, the JSPS Institutional Program for Young Researchers at Keio University, and the Uehara Memorial Foundation, Japan. J.E.E.O. is supported by CIHR, IBBME International Scholars Program at University of Toronto, and Gålöstiftelsen, Sweden. E.P. is supported by RESTRACOMP Research Fellowship from SickKids. P.W.Z. is supported by the Canada Research Chair in Stem Cell Bioengineering. We thank J. Chenoweth and R. McKay for RNA-seq data on EpiSCs/mESCs, B. MacArthur, I. Lemischka, K. Hayashi, A. Surani for single-cell data on mESCs; G. Martello and A. Smith for knock-down data on mESCs; C. Yoon for providing mesoderm progenitor cells; Dr. M. Nakanishi for initiating chimera analysis; and Dr. C. Bauwens for the critical reading and editing of this manuscript.

### Author contributions

AY-K designed and performed *in silico* study and bioinformatics analysis. KO designed and performed *in vitro* experiments. JO performed *in vitro* experiments and bioinformatics analysis. MAL implemented software for the *Garuda* platform. EP designed and performed, and JR supervised *in vivo* experiments. AY-K, KO, and PWZ designed the project and AY-K, KO, JO, MAL, and PWZ wrote the manuscript.

### Conflict of interest

The authors declare that they have no conflict of interest.

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
