## [Review Process File · Molecular Systems Biology]

Modeling signaling-dependent pluripotency with boolean logic to predict cell fate transitions

Ayako Yachie-Kinoshita, Kento Onishi, Joel Ostblom, Matthew A. Langley, Eszter Posfai, Janet Rossant and Peter W. Zandstra

Review timeline:

Submission date:	24 August 2017
Editorial Decision:	5 October 2017
Revision received:	21 November 2017
Accepted:	20 December 2017

Editor: Maria Polychrnoidou

Transaction Report:

1st Editorial Decision

5 October 2017

Thank you again for submitting your work to Molecular Systems Biology. We have now heard back from the three referees who agreed to evaluate your study. As you will see below, the reviewers raise a series of concerns, which we would ask you to address in a revision of the manuscript.

The reviewers' recommendations are rather clear so I think that there is no need to repeat the points listed below. If there is any point you would like to discuss in further detail please feel free to contact me.

REVIEWER REPORTS

Reviewer #1:

Yachie-Kinoshita et al. introduce an asynchronous Boolean framework to describe heterogeneity in pluripotent embryonic stem cells. This is an interesting development as heterogeneity is generally ignored when building GRNs. However, it is not clear whether the validations of the model are truly independent of the data used to build the model. In the published microarray data the authors used, many perturbations are used (and therefore feed in the model) and overlap with the ones used by the authors. The authors predict culture conditions enabling the generation of Cdx2+ cells. The characterization of these Cdx2+ cells falls short of demonstrating that it is a physiologically relevant population. It would be very interesting if the authors could demonstrate if their framework could be used to find conditions that improve differentiation efficiencies. With the susceptibility and sustainability concepts at hand, this should be feasible. Making the modeling framework available to the community is commendable.

Major concerns:

- The robustness of the GRN inference to the number of datasets used is not discussed. Indeed, the authors use a very large number of datasets (1295) to generate the GRN. What is the minimal number of datasets necessary to build a GRN that behaves like the one described? More importantly, how many different experimental conditions are necessary? For example, is it possible to predict the effects of ERK inhibition without using any experimental data that uses ERK inhibition?
- The results of chimera contribution experiments are not convincing statistically due to the small number of assessed chimeras and the fact that the two tested cell lines do not show consistent results. In addition, The use of feeders to maintain the cells used for chimera assays is a confounding factor due to all the growth factors secreted by the feeder themselves. Feeder-free cells should be used. In the plot Figure EV4d, there is no difference between the lineage contribution of 2iL and 2iJ. There is a discrepancy between the plot and the table of Figure EV4d: the fractions in the plot do not match the numbers in the table.
- The actual GRN inferred by the authors should be displayed as a graph in addition to the table provided in supplements. This will improve readability.
- In order to better characterize the Cdx2+ cells generated under 2iJ+B-A conditions, the authors should use a fluorescent reporter of Cdx2 expression and isolate by FACS the Cdx2-expressing cells.
- The sentence lines 42-44 page 7 is misleading as reference 52 cited by the authors describes the generation of trophoblast cells from mESCs only using media components (including BMP4).

Minor concerns:

- Strongly Connected Components associated with different pluripotency conditions should be displayed in supplements.
- in Figure 4i, it is hard to judge the locations of the GFP positive cells. Individual stacks should be displayed instead of a maximum projection of the entire embryo.
- in Figure 2c, the shading is very hard to distinguish.
- in the last paragraph of section 3.1, it is superfluous to display 2 significant figures for pvalues in the range of 10^{-270} .

Reviewer #2:

This is an elegant and extensive study utilizing previously published and new gene expression datasets on mouse ESCs expanded in a variety of naive and primed pluripotent state, and used to build a model inferring GRNs and susceptibility to stimuli in different conditions and upon exposure to different cytokines.

The authors used boolean logic and developed unique codes to build and validate different models and networks that control these different pluripotent states.

The conclusions drawn by the study are valid and well supported by the data. They are also consistent with previous work Dunn et al. Science 2014 where relevant, and in fact, this study is much more extensive and addresses many other novel aspects and conditions.

The data on using JAK inhibitor to promote cells towards trophoblast lineage is novel and back by "wet" assays in vitro and in vivo.

I do not see any major or minor flaws or caveats in this study, and thus i have no requests when revising or publishing this manuscript.

Reviewer #3:

The authors created a boolean logic model of a mouse Embryonic Stem Cell (mESC) gene regulatory network (GRN) of 29 genes and 7 signaling pathway intermediates and then applied asynchronous updates, starting at random initial states, to simulate the trajectories of the GRN and predict how certain extrinsic cues could affect the maintenance of pluripotency in an mESC population. Unlike previous boolean models of pluripotency, here the authors quantify population heterogeneity and identify subpopulations that emerge by analyzing the strongly connected components (SCCs) of the simulation. Their model was validated by comparing the *in silico* gene expression levels to *in vitro* experiments upon addition of select extrinsic cues to the cell culture medium. The authors then quantify new metrics of population state in the form of "sustainability" and "susceptibility" to describe the combinatorial effect of LIF and 2i (inhibition of MEK and GSK3-beta) on the destabilization of mESC pluripotency in response to extrinsic cues. Finally, the authors used the model to predict conditions that would induce trophectodermal fates *in vitro*; however, they found using a morula aggregation chimera assay that priming mESCs with inhibitors of MEK, GSK3beta, and LIF is not sufficient to induce trophectodermal commitment *in vivo*.

The model development and analysis are rigorous with proper validation experiments and performance tests in the Supplementary Notes. The authors provide a technical advance in the logic modeling of PSC population heterogeneity in response to combinatorial extrinsic stimuli by applying R-ABS and analyzing the SCCs. The work also quantifies fundamental traits of a PSC GRN (the susceptibility and sustainability) in a novel way. The model's code is available to the public, and the computational biology community might be interested in studying and adapting it to additional development and differentiation questions. The Supplementary Notes are presented very clearly and thoroughly.

Major points:

1. A discussion on the limitations of a boolean network models would be appropriate. The model neglects the dose-dependent relationship between TFs and target genes, as well as assumes linearity among all regulatory relationships by using the Pearson's correlation. How might these limitations affect the predictive power of the model?
2. Page 3, Line 13, Section "Simulation Framework for PSCs": is this a novel result? Asynchronous boolean models have been applied to GRNs previously and the authors apply a publicly available software to run the ABS (BooleanNet). Strongly Connected Components are an established concept in graph theory. A could could be made for putting this in the METHODS section or Supplementary Notes instead of RESULTS.
- Page 3, Line 43, Section "Mouse ESC GRN Construction" Is this a novel result? GGM has been applied to gene regulatory networks previously. This seems to fit better in METHODS section or Supplementary Notes instead of RESULTS.

Minor points:

1. The flow of the paper could be re-ordered to introduce GRN construction before boolean modeling framework to follow a more logical progression.
2. Page 3, Line 9: remove "the" from "and the heterogeneity"
3. Page 4, Line 32, 33, 34 and 38: The number of regulatory interactions between genes in the GRN is inconsistent, is it 105 or 95 total? (95 from GGM inference + 10 from model fitting OR 85 from GGM inference + 10 from model fitting) Table M1 shows 95 from GGM inference + 10 from model fitting = 105 total
4. Figure 2a and 2b: why are these grouped into the same figure when they are referenced in completely different sections of the paper?
5. Figure 3: subplot ordering is not systematic, swap b and c subplot placement
6. Page 7, Line 31: neither main text nor captions explicitly mention use of CH and PD (shown in Figure 3e) as agonists
7. Figure 3f: why is "2i-L+B-A" in each plot? The caption says that the color of the data point corresponds to the media condition of the data point
8. Figure EV3a: y-axis details? How did you process the immunocytochemistry data to get that

scale?

9. Supplementary Notes Section 1-4: better placed within Section 4

10. Supplementary Notes Figure M2.1: shows the median value was calculated, whereas the text above says the "mean values of probes with the same gene annotation"

11. Supplementary Notes Figure M2.1: can remove "every rest of" from sentence above pcor equation

Supplementary Notes Figure M2.1: shows that "pValue of calculation < 0.05 " whereas the text below says "(2) the p-value was greater than 0.05"

1st Revision - authors' response

21 November 2017

Reviewer #1:

Yachie-Kinoshita et al. introduce an asynchronous Boolean framework to describe heterogeneity in pluripotent embryonic stem cells. This is an interesting development as heterogeneity is generally ignored when building GRNs. However, it is not clear whether the validations of the model are truly independent of the data used to build the model. In the published microarray data the authors used, many perturbations are used (and therefore feed in the model) and overlap with the ones used by the authors. The authors predict culture conditions enabling the generation of Cdx2+ cells. The characterization of these Cdx2+ cells falls short of demonstrating that it is a physiologically relevant population. It would be very interesting if the authors could demonstrate if their framework could be used to find conditions that improve differentiation efficiencies. With the susceptibility and sustainability concepts at hand, this should be feasible. Making the modeling framework available to the community is commendable.

We appreciate the reviewer's comments and summary of our approach. The reviewer raises two important points: whether the datasets used for model construction (i.e. GRN inference) versus model validation are independent, and whether our framework can feasibly identify conditions that improve differentiation efficiencies. A detailed response to the first concern raised by the reviewer is outlined below in our point-by-point responses. Toward the second point, our model is developed around quantitatively predicting exit/loss of pluripotency—as captured by the sustainability and susceptibility scores—but not how cells progress further along each differentiation path. As such, our model is capable of predicting conditions that give rise to pluripotent states which are more receptive to differentiation signals; for example, we predicted and experimentally validated that cells in the 2iJ (Jaki) condition are more responsive to Bmp and Activin A/Nodal signaling. However, the model in its current form does not make predictions about the efficiency with which these cells would progress toward specific differentiated lineages. Whether our modelling strategy can be extended to predict refined conditions for differentiating cell populations is an intriguing topic, and one we intend to pursue in future studies.

Major concerns:

1. The robustness of the GRN inference to the number of datasets used is not discussed. Indeed, the authors use a very large number of datasets (1295) to generate the GRN. What is the minimal number of datasets necessary to build a GRN that behaves like the one described? More importantly, how many different experimental conditions are necessary? For example, is it possible to predict the effects of ERK inhibition without using any experimental data that uses ERK inhibition?

We thank the reviewer for this point regarding the data used for model construction. To address the robustness of our GRN inference approach to the datasets, we tested for batch effects in the datasets by randomly dividing the datasets into two groups that can then be compared with respect to the predicted strength of gene-to-gene connectivity. From this, we found that batch effects among the two randomly divided datasets are negligible for inference of the GRN, thus demonstrating the robustness of the GRN inference to the dataset. (please see the Appendix 2-2 and Figure S2.2a).

In the revised manuscript, we further assess the robustness of our inference method to the number of microarray datasets used. We compared the gene-to-gene relationships inferred by CLR using the full set of microarray data (i.e. all 1,295 samples) to those inferred from variously-sized partial datasets of randomly-selected microarray profiles (please see **Figure R1 below**). Surprisingly, around 200 samples were sufficient to replicate the gene-to-gene relationships inferred from the full set of microarray samples.

We added the following sentence to clarify the robustness of our inference method:

“To further characterize the robustness of GRN inference to the input microarray data, we compared the gene-gene relationships inferred by CLR using the full set of microarray data (1,295 samples) to those inferred from variously-sized partial datasets from of randomly selected microarray profiles (Figure S2.2b). The results indicated that a relatively small number of samples (>200) are feasible to replicate the full dataset showing correlation coefficients as high as 0.9, and sufficient for robust GRN inference.” (Appendix 2-2, page 8).

Figure R1. Assessment of the robustness of the expression dataset. (replicated from Appendix Figure S2.2b) Small dots represent Pearson's correlation coefficient between gene-to-gene relationships inferred by CLR algorithm based on either the full (1,295) dataset or the partial dataset where the number of samples is indicated in x-axis.

Our modeling approach consists of three major steps: 1) Manual curation-based selection of the model components (e.g. genes and signals) and definition of the regulatory relationships among them; 2) GRN inference using microarray datasets to identify potential gene regulations followed by curation to assign directionalities; and 3) Simulation-based model selection to refine the inferred but not confirmed relationships. It is challenging to distinguish the data source that leads to a specific phenotype in the model. On the other hand, as has been emphasized by others, input dataset that containing a wide range of information on the underlying GRN structure may have advantages in reverse engineering (i.e. GRN inference) rather than systematic perturbations (Bhosale et al., 2013).

2. The results of chimera contribution experiments are not convincing statistically due to the small number of assessed chimeras and the fact that the two tested cell lines do not show consistent results. In addition, the use of feeders to maintain the cells used for chimera assays is a confounding factor due to all the growth factors secreted by the feeder themselves. Feeder-free cells should be used. In the plot Figure EV4d, there is no difference between the lineage contribution of 2iL and 2iJ. There is a discrepancy between the plot and the table of Figure EV4d: the fractions in the plot do not match the numbers in the table.

We appreciate the reviewer's comments and concerns with respect to the chimera experiments. With respect to the statistical significance of the chimera studies, the number of chimeras examined here are well within the typical range for studies looking at chimeric embryos (Morgani et al., 2013; Macfarlan et al., 2012). It is also well-established that different ES cell lines from different strains and passage numbers (and even different clones from the same line) show variability in chimera contributions. Therefore, the differences in behavior between cell lines is not unexpected. It is important to note, however, that both cell lines show the same trend – a higher ratio of chimeras with cells in TE positions in 2iJ conditions compared with the 2iL conditions.

Related to the chimera data, we corrected the error in the table in former Fig.EV4d (Fig. EV5d in current manuscript) from “14, 8, 11, 7” to “8, 14, 7, 11” for “2iL-TE, 2iL-EPI, 2iJ-TE, 2iJ-EPI”, respectively. We appreciate this reviewer for pointing out this mistake.

As for the use of feeders, there are two main reasons why we chose feeder/serum-containing conditions for our *in vivo* and *in vitro* experiments. First, as the reviewer pointed out, the presence of feeders/serum enhances cell survival in the presence of potent differentiation signals and therefore aided downstream cell fate characterization in these experiments. Second, we have seen clear effects from the addition of exogenous BMP4 and Activin A on serum-cultured mESCs *in vitro*. For the *in vitro* analysis, we examined OSN expression in different signal combinations by high content screening and confirmed the observed effects in the absence of serum (Extended View Figure EV4h).

3. *The actual GRN inferred by the authors should be displayed as a graph in addition to the table provided in supplements. This will improve readability.*

Thanks to the reviewer’s suggestion, a graph of the predicted and curated network in the model has been developed and is displayed in Expanded View Figure EV2.

Figure R2. Assessment of the robustness of the expression dataset. (replicated from Expanded View Figure EV2) A network view of the mESC-GRN model where rectangles indicate model components including genes (white and green for genes with and without outgoing regulatory interactions), signaling activities (gray), and cytokines or small molecule inputs (dark gray). Edges between rectangles represent the regulatory relationships between genes (solid lines) and within signaling pathways (dotted lines). The edge color indicates either literature curation-based (black) or inferred/predicted (red) regulations.

4. *In order to better characterize the Cdx2+ cells generated under 2iJ+B-A conditions, the authors should use a fluorescent reporter of Cdx2 expression and isolate by FACS the Cdx2-expressing cells.*

We had attempted such experiments using nuclear and cytoplasmic Cdx2 reporter lines but the Cdx2 reporter was too weak to reliably sort positive cells. We also generated Cdx2-eGFP homozygous TS cells as a gating control, but the reporter was too weak in these efforts as well. We will develop other reporter selection system in future work.

5. *The sentence lines 42-44 page 7 is misleading as reference 52 cited by the authors describes the generation of trophoblast cells from mESCs only using media components (including BMP4).*

Thanks to the reviewer for pointing this out. We removed 'but not typically' from the sentence: "Differentiation of naïve mESCs to trophoblast stem (TS) cell-like cells occurs upon the forced expression of the trophoblast master regulator Cdx2, through the addition of medium components (Hayashi et al., 2010; Niwa et al., 2005). Moreover, apparent totipotency from mESCs derived in 2i (Morgani et al., 2013) has been reported, and BMP signal activation helps drive trophoblast gene expression from mouse and human primed PSCs (Bernardo et al., 2011; Brons et al., 2007; Vallier et al., 2009)." (Page 8 Line 11-16)

Minor concerns:

6. *Strongly Connected Components associated with different pluripotency conditions should be displayed in supplements.*

We agree with the reviewer's suggestion, and the condition-dependent populations of stem cells that correspond to the SCCs in the state transition have been displayed in Figure 3b.

Figure R3. Condition-dependent pluripotent cell populations correspond to Strongly Connected Components (SCCs) in the state transition graphs of asynchronously updated Boolean models. (replicated from Figure 3b) Gray dots represent unique profiles and edges represent state transitions among the profiles. Colored edges indicate the transitions within population-specific SCCs. The number of simulations and the number of steps in each simulation were 300-100, 300-100, 300-300 for LS, 2iL and bF+A condition, respectively.

7. *in Figure 4i, it is hard to judge the locations of the GFP positive cells. Individual stacks should be displayed instead of a maximum projection of the entire embryo.*

According to the reviewer's suggestion, we have displayed a single plane image in the revised Figure 5i.

8. *in Figure 2c, the shading is very hard to distinguish.*

In the new version of Figure 3c, we changed the color-coding of the line tracing the shaded region.

9. *in the last paragraph of section 3.1, it is superfluous to display 2 significant figures*

for p-values in the range of 10^{-270} .

We revised the sentence as follows: “*Note that other TFs reported to have important roles for pluripotency maintenance such as Tcfcp2l1 and Klf5 were not included due to significant overlaps in correlated gene partners with Esrrb and Klf4 (p-values < 10^{-270} for positively correlated genes), respectively.*”

References:

- Bhosale, R., Jewell, J.B., Hollunder, J., Koo, A.J.K., Vuylsteke, M., Michael, T., Hilson, P., Goossens, A., Howe, G.A., Browse, J., Maere, S., 2013. Predicting Gene Function from Uncontrolled Expression Variation among Individual Wild-Type Arabidopsis Plants. *Plant Cell* tpc.113.112268.
- Morgani, S.M., Canham, M.A., Nichols, J., Sharov, A.A., Migueles, R.P., Ko, M.S.H., Brickman, J.M., 2013. Totipotent embryonic stem cells arise in ground-state culture conditions. *Cell Rep.* 3, 1945–1957.
- Macfarlan, T.S., Gifford, W.D., Driscoll, S., Lettieri, K., Rowe, H.M., Bonanomi, D., Firth, A., Singer, O., Trono, D., Pfaff, S.L., 2012. Embryonic stem cell potency fluctuates with endogenous retrovirus activity. *Nature* 487, 57–63.

Reviewer #2:

This is an elegant and extensive study utilizing previously published and new gene expression datasets on mouse ESCs expanded in a variety of naive and primed pluripotent state, and used to build a model inferring GRNs and susceptibility to stimuli in different conditions and upon exposure to different cytokines.

The authors used boolean logic and developed unique codes to build and validate different models and networks that control these different pluripotent states.

*The conclusions drawn by the study are valid and well supported by the data. **They are also consistent with previous work Dunn et al. Science 2014 where relevant, and in fact, this study is much more extensive and addresses many other novel aspects and conditions.***

The data on using JAK inhibitor to promote cells towards trophoblast lineage is novel and back by "wet" assays in vitro and in vivo.

I do not see any major or minor flaws or caveats in this study, and thus i have no requests when revising or publishing this manuscript.

We are very thankful to this reviewer for this overview and appreciation of the aims and approach of our manuscript.

Reviewer #3:

The authors created a boolean logic model of a mouse Embryonic Stem Cell (mESC) gene regulatory network (GRN) of 29 genes and 7 signaling pathway intermediates and then applied asynchronous updates, starting at random initial states, to simulate the trajectories of the GRN and predict how certain extrinsic cues could affect the maintenance of pluripotency in an mESC population. Unlike previous boolean models of pluripotency, here the authors quantify population heterogeneity and identify subpopulations that emerge by analyzing the strongly connected components (SCCs) of the simulation. Their model was validated by comparing the in silico gene expression levels to in vitro experiments upon addition of select extrinsic cues to the cell culture medium. The authors then quantify new metrics of population state in the form of "sustainability" and "susceptibility" to describe the combinatorial effect of LIF and 2i (inhibition of MEK and GSK3-beta) on the destabilization of mESC pluripotency in response to extrinsic cues. Finally, the authors used the model to predict conditions that would induce trophectodermal fates in vitro; however, they found using a morula aggregation chimera assay that priming mESCs with inhibitors of MEK, GSK3beta, and LIF is not sufficient to induce trophectodermal commitment in vivo.

The model development and analysis are rigorous with proper validation experiments and performance tests in the Appendix. The authors provide a technical advance in the logic modeling of PSC population heterogeneity in response to combinatorial extrinsic stimuli by applying R-ABS and analyzing the SCCs. The work also quantifies fundamental traits of a PSC GRN (the susceptibility and sustainability) in a novel way. The model's code is available to the public, and the computational biology community might be interested in studying and adapting it to additional development and differentiation questions. The Appendix are presented very clearly and thoroughly.

We appreciate this reviewer's summary and thorough review of our manuscript. We are also very pleased that the reviewer appreciates the novelty of the fundamental metrics for PSC GRN we have proposed.

Major points:

1. A discussion on the limitations of a boolean network models would be appropriate. The model neglects the dose-dependent relationship between TFs and target genes, as well as assumes linearity among all regulatory relationships by using the Pearson's correlation. How might these limitations affect the predictive power of the model?

The reviewer correctly identifies important limitations given the assumptions of Boolean network modeling with respect to dose-dependency and non-linearity. Boolean models assume that target genes respond to transcription factor doses in a switch-like fashion. This mathematical simplification is analogous to a steep Hill function (commonly used in kinetic differential equation models of gene regulation) with high cooperativity. Dose-dependent responses of some target genes to TF activity has been reported in some biological systems, and such cases can be simulated by relaxing the Boolean assumption (for example, to include fuzzy logic or Petri nets). In contrast, however, many studies have demonstrated that gene expression is bimodality distributed at the single cell level; for example, in massive single cell RNA-seq on immune cells (Shalek et al., 2013) and in single cell qPCR analysis of mESCs (MacArthur et al., 2012; Xu et al., 2014). Given these observations, we opted for the simpler Boolean assumption in our model. Indeed, our model suggests that many aspects of PSC fate behavior can be adequately captured computationally without considering dose-dependent effects.

Importantly, the random asynchronous Boolean modeling approach we employ in this work avoids some limiting assumptions of existing Boolean simulation approaches. While previous studies assume that observed phenotypes must correspond to individual steady state profiles in the Boolean state transition graph, we broadened our focus to include

strongly connected components (SCCs) as an analog to PSC populations composed of single cells in dynamic yet stable heterogeneity. By averaging over all states in an SCC, we can calculate a non-binary average expression profile for each simulated PSC condition and directly compare against continuously-valued experimental gene expression data. Thus, our model's assumption of binary gene expression at the single cell level does not preclude non-binary gene expression arising from heterogeneity at the population level.

To emphasize these points, we inserted the following text in the manuscript:

"We believe that a binarized representation of gene expression, which is a common simplification for Boolean-based simulations, is relevant at the single cell level given the accumulated observations of bimodal distributions in single cell gene expression profiles in mESCs (MacArthur et al., 2012; Xu et al., 2014) and in other cell types (Shalek et al., 2013)."
(Page 3 Lines 4-8)

2. Page 3, Line 13, Section "Simulation Framework for PSCs": is this a novel result? Asynchronous boolean models have been applied to GRNs previously and the authors apply a publicly available software to run the ABS (BooleanNet). Strongly Connected Components are an established concept in graph theory. A could be made for putting this in the METHODS section or Appendix instead of RESULTS.

A key novelty of our simulation approach is how we define a cell population and calculate its average gene expression levels. This strategy differs from other Asynchronous Boolean simulations because we define the border of pluripotent cell populations on the basis of the dynamic attractor states of different single cell profiles (SCCs). By defining a population as a stable ensemble of single cell states, we can predict the continuous-valued population average expression levels—something which to date has not been successfully captured by existing Boolean simulations or static GRN analysis. Given this, we believe that the simulation framework described in this section is a key result of our study.

3. Page 3, Line 43, Section "Mouse ESC GRN Construction" Is this a novel result? GGM has been applied to gene regulatory networks previously. This seems to fit better in METHODS section or Appendix instead of RESULTS.

Our modeling approach consists of three steps each of which is indispensable: 1) Manual curation-based selection of the model components (e.g. genes and signaling) and definition of the regulatory relationships among them; 2) GRN inference using a microarray dataset model to include potential gene regulations and curation-based definition of their directionalities; and 3) Simulation-based model selection to define the inferred but not confirmed relationships. Although the strategy involved in each individual step is established, the full procedure is quite unique and needs attention; thus, we believe the modeling procedure is better placed in the RESULTS section. Accordingly, the GRN inference (through iteration of GGM) itself and the details of model selection are in the METHODS section.

Minor points:

1. The flow of the paper could be re-ordered to introduce GRN construction before boolean modeling framework to follow a more logical progression.

This study introduced a novel framework for a signal-GRN network, which was then applied to the mESC-GRN through GRN construction. We agree with the reviewer that the flow was not clear. In the revised manuscript, the section of GRN construction starts with: *"Next, we applied the proposed simulation framework to mESC-GRN. To build the model we first ..."*
(Page 4 Line 8)

2. Page 3, Line 9: remove "the" from "and the heterogeneity"

We refined the sentence accordingly. We thank the reviewer for the careful reading.

3. Page 4, Line 32, 33, 34 and 38: *The number of regulatory interactions between genes in the GRN is inconsistent, is it 105 or 95 total? (95 from GGM inference + 10 from model fitting OR 85 from GGM inference + 10 from model fitting) Table M1 shows 95 from GGM inference + 10 from model fitting = 105 total*

Thanks to the reviewer's question, we noticed and fixed the incorrect numbers. The total number of gene-to-gene relationships (=105) includes 19 known self-activations and 86 inferred links. The directionalities of the ten links out of 86 were determined by model fitting. To avoid the confusion, Appendix Table M1 is divided into two groups (a. Inferred gene-to-gene relationships, and b. Known self-activations).

The corresponding sentences from the manuscript are as follows:

- "**19** regulations encompassing double positive or double negative regulatory circuits and known self-activations for seven genes (Appendix Table M1b)" (main text, page 4 Line 17)
- "The network included **86** inferred pairwise gene regulatory relationships (Appendix Table M1a). Directionality was determined for **76** of these gene pairs by either experimental evidence or gene function annotation. The directionality for the remaining **10** gene pairs was determined by subsequent model selection based on fitting to reported single cell gene expression frequency." (main text, page 4 Lines 40-44)
- "As shown in Table M1a, the directionalities of **76** out of **86** inferred gene-gene regulatory links were determined in this evidence-based step." (The last paragraph in Appendix 3-2)

4. Figure 2a and 2b: *why are these grouped into the same figure when they are referenced in completely different sections of the paper?*

We agree with the reviewer's comment, and moved former Figure 2b-d into Figure 3. In the revised manuscript, we included visualized Strongly Connected Components associated with different pluripotency conditions in Figure 3b.

Figure R3. Condition-dependent pluripotent cell populations correspond to Strongly Connected Components (SCCs) in the state transition graphs of asynchronously updated Boolean models. (replicated from Figure 3b) Gray dots represent unique profiles and edges represent state transitions among the profiles. Colored edges indicate the transitions within population-specific SCCs. The number of simulations and the number of steps in each simulation were 300-100, 300-100, 300-300 for LS, 2iL and bF+A condition, respectively.

5. *Figure 3: subplot ordering is not systematic, swap b and c subplot placement*

We replaced the positions of the subplots according to the reviewer's suggestion (Figure 4b and 4c).

6. *Page 7, Line 31: neither main text nor captions explicitly mention use of CH and PD (shown in Figure 3e) as agonists*

To address the reviewer's concern, we define the agonists in the Figure legend where the 2i condition was first referenced: "*The 2i condition consists of CHIR99021(CH) and PD0325901(PD).*" (Figure legend for Fig.4a)

7. *Figure 3f: why is "2i-L+B-A" in each plot? The caption says that the color of the data point corresponds to the media condition of the data point*

With the labels of "2i-L+B-A", we highlight the specific condition which showed lower expressions of Oct4/Sox2/Nanog in spite of the existence of components of 2i - whereas the colors (red, blue, orange and white) are corresponding to the four medium conditions depending on the LIF-JAK/STAT and Wnt- β -catenin pathway manipulations (+LIF+iGSK3 β , +JAKi+iGSK3 β , +LIF+DKK1, +JAKi+DKK1, respectively).

8. *Figure EV3a: y-axis details? How did you process the immunocytochemistry data to get that scale?*

The frequencies of positive cells for single genes were assessed by counting the single cells whose expression levels are above a certain threshold (assessed based on the bimodal distribution of the expression level in LIF+Serum conditions) which is common across each technical replicate (i.e. each plate) for each of the two biological replicates run for each condition. We clarified this in "*In vitro* Immunostaining and Quantification" in the METHODS section. In Figure EV4a (Figure EV3a in former version), we defined and showed the summation of the frequencies of Oct4, Sox2 and Nanog (OSN levels) as an indicator of pluripotency.

9. *Appendix Section 1-4: better placed within Section 4*

As the metric of sustainability is used to calculate average expression and in the following subpopulation analysis, the information supplied in Appendix section 1-4 must be presented before sections 1-5 and 1-6. To improve the flow, we inserted the following sentence in the Section 1-4, before the mathematical detail of the sustainability score: "*This score can be used to estimate the stability of each SCC which reflects the intrinsic stability of the GRN over time in the absence of extrinsic perturbations (see Section 4. "Characterization of PSCs via pluripotency, sustainability and susceptibility").*"

10. *Appendix Figure S2.1: shows the median value was calculated, whereas the text above says the "mean values of probes with the same gene annotation".*

Thanks to the reviewer's careful review we have revised "median" to "mean" in Figure S2.1.

11. *Appendix Figure S2.1: can remove "every rest of" from sentence above pcor equation.*

These words have been removed according to the reviewer's suggestion.

12. Appendix Figure S2.1: shows that "pValue of calculation < 0.05" whereas the text below says "(2) the p-value was greater than 0.05"

Thanks to the reviewer's attentive edit, the error was fixed from "greater than" to "less than".

References:

- Shalek, A.K., Satija, R., Adiconis, X., Gertner, R.S., Gaublomme, J.T., Raychowdhury, R., Schwartz, S., Yosef, N., Malboeuf, C., Lu, D., Trombetta, J.J., Gennert, D., Gnirke, A., Goren, A., Hacohen, N., Levin, J.Z., Park, H., Regev, A., 2013. Single-cell transcriptomics reveals bimodality in expression and splicing in immune cells. *Nature* 498, 236–240.
- MacArthur, B.D., Sevilla, A., Lenz, M., Müller, F.-J., Schuldts, B.M., Schuppert, A.A., Ridden, S.J., Stumpf, P.S., Fidalgo, M., Ma'ayan, A., Wang, J., Lemischka, I.R., 2012. Nanog-dependent feedback loops regulate murine embryonic stem cell heterogeneity. *Nat. Cell Biol.* 14, 1139–1147.
- Xu, H., Ang, Y.-S., Sevilla, A., Lemischka, I.R., Ma'ayan, A., 2014. Construction and validation of a regulatory network for pluripotency and self-renewal of mouse embryonic stem cells. *PLoS Comput. Biol.* 10, e1003777.

Thank you again for sending us your revised manuscript. We have now heard back from the two reviewers who were asked to evaluate your study. As you will see below, the reviewers are satisfied with the modification made and think that the study is suitable for publication. As such, I am pleased to inform you that your paper has been accepted for publication.

REVIEWER REPORTS

Reviewer #1:

In this revision, the authors have answered satisfactorily my concerns as well as Reviewer 3's.

Reviewer #3:

The authors have addressed all concerns of this reviewer.

Corresponding Author Name: Peter W. Zandstra

Manuscript Number: MSB-17-7952